# GENEX: GENERATING AN EXPLORABLE WORLD

**Taiming Lu, Tianmin Shu, Alan Yuille, Daniel Khashabi, Jieneng Chen**
Johns Hopkins University
jchen293@jhu.edu

## ABSTRACT

Understanding, navigating, and exploring the 3D physical real world has long been a central challenge in the development of artificial intelligence. In this work, we take a step toward this goal by introducing *GenEx*, a system capable of planning complex embodied world exploration, guided by its generative imagination that forms expectations about the surrounding environments. *GenEx* generates high-quality, continuous 360-degree virtual environments, achieving high loop consistency and active 3D mapping over extended trajectories. Leveraging generative imagination, GPT-assisted agents can undertake complex embodied tasks, including goal-agnostic exploration and goal-driven navigation. Agents utilize imagined observations to update their beliefs, simulate potential outcomes, and enhance their decision-making. Training on the synthetic urban dataset *GenEx-DB* and evaluation on *GenEx-EQA* demonstrate that our approach significantly improves agents' planning capabilities, providing a transformative platform toward intelligent, imaginative embodied exploration.

| | Website | https://www.GenEx.world/ |
|---|---|---|
| | Code | https://github.com/Beckschen/GenEx |
| | ArXiv | https://arxiv.org/abs/2412.09624 |

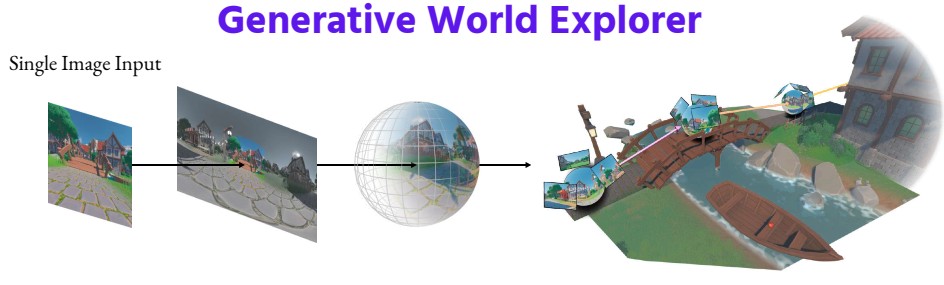

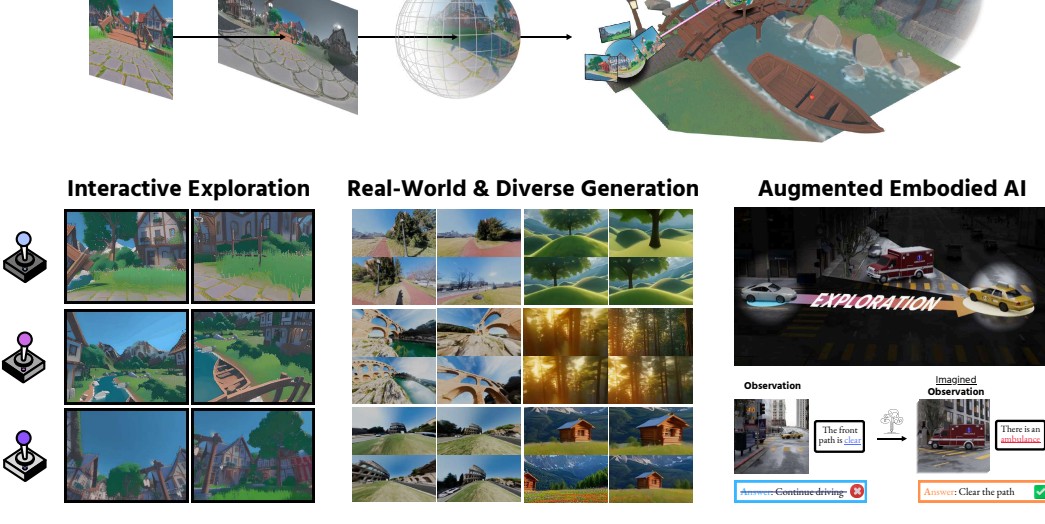

Figure 1: *GenEx* explores an imaginative world, created from a single RGB image and brought to life as a generated video (top). With interactive and diverse generation, *GenEx* enables an agent to *imaginatively* explore a large-scale 3D world and acquire imagined observation to augment embodied intelligence (bottom).

# 1 INTRODUCTION

Humans navigate and interact with the three-dimensional world by perceiving their surroundings, taking actions, and engaging with others. Through these interactions, they form *mental models* to simulate the world (Johnson-Laird, 1983). These models allow for internal representations of reality, aiding reasoning, problem-solving, and prediction through language and imagery.

In parallel, this understanding of natural intelligence has inspired the development of artificial intelligence systems that create computational analogs of mental models (Ha & Schmidhuber, 2018; LeCun, 2022; Diester et al., 2024). These *world models* (WMs) (Ha & Schmidhuber, 2018; LeCun, 2022) aim to mimic human understanding and interaction by predicting future world states (e.g., the existence, properties and location of the objects in a scene) to help agents make informed decisions. Recently, generative vision models (Ho et al., 2020; OpenAI, 2024; Bai et al., 2024) have increased interest in developing world models for predictive simulation of the world (Du et al., 2024a; Yang et al., 2024b;c; Wang et al., 2024a). However, these works focus solely on state transition probabilities without explicitly modeling agents' observations and beliefs. Explicitly modeling observation and belief is crucial because we often deal with partially observable environments where the true world state is unknown. An embodied agent is inherently a POMDP agent (Kaelbling et al., 1998): instead of full observation, the agent has only partial observations of the environment. To make rational decisions, the agent must form a belief, an estimate of the environment it is currently in. This belief may be incomplete or biased, but it can be revised through incoming observations obtained by physically exploring the environment.

Typically, in an unfamiliar environment, an embodied agent must acquire new observations through physical exploration to understand its surroundings, which is inevitably costly, unsafe, and time-consuming. However, if the agent can *imagine* hidden views by mentally simulating exploration, it can update its beliefs without physical effort. This enables the agent to take more informed actions and make more robust decisions. Consider the scenario in Fig. 1, suppose you are approaching an intersection. The light ahead is green, but you suddenly notice that the yellow taxi in front has come to an abrupt, unexpected stop. A surge of confusion and anxiety hits you, leaving you uncertain about the reason behind its halt. Physically investigating the situation would be unsafe and even impossible at that moment. However, by standing in the taxi's position in your own imagination and envisioning the surroundings from its perspective, you sense a possible motivation behind the taxi's puzzling behavior: *perhaps an ambulance is approaching*. Consequently, you *clear the path for the emergency vehicle*, a timely and decisive choice, thanks to your imagination.

To build agents capable of imaginative exploration in a physical world, we propose ***Gen**erative World **Ex**plorer (GenEx)*, a video generative model that conditions on the agent's current egocentric (first-person) view, incorporates intended movement direction as an action input, and generates future egocentric observation. Although prior works (Tewari et al., 2023) can render novel views of a scene based on 3D models (Yu et al., 2021), the limited render distance and the limited field of view (FOV) constrain the range and coherence of the generated video. Fortunately, video generation offers the potential to extend the exploration range. To address the FOV constraint, we utilize panoramic representations to train our video diffusion models with spherical-consistent learning. As a result, the proposed *GenEx* model achieves impressive generation quality while maintaining coherence and 3D consistency throughout long-distance exploration.

Furthermore, the proposed *GenEx* can be applied to the embodied decision making. With *GenEx*, the agent is able to imagine hidden views via imaginative exploration, and revise its belief. The revised belief allows the agent to take more informed actions. Technically, we define the agent's behavior as an extension of POMDP with *imagination-driven belief revision*. Notably, the proposed *GenEx* can naturally be extended to multi-agent scenarios, where one agent can mentally navigate to the positions of other agents and update its own beliefs based on imagined beliefs of the other agents.

In summary, our key contribution is three-fold:

- We introduce *GenEx*, a novel framework that enables agents to imaginatively explore the world with high generation quality and exploration consistency.
- We present one of the first approaches to integrate generative video into the partially observable decision process by introducing the imagination-driven belief revision.
- We highlight the compelling applications of *GenEx*, including multi-agent decision-making.

## 2    RELATED WORKS

**Generative video modeling.**    Diffusion models (DMs) (Sohl-Dickstein et al., 2015; Ho et al., 2020) have proven effective in image generation. To render high-resolution images, the latent diffusion models (LDMs) (Rombach et al., 2022) are proposed to denoise in the latent space. Similarly, video diffusion models (Blattmann et al., 2023b; Wang et al., 2023a; Blattmann et al., 2023a; Song et al., 2025) use VAE models to encode video frames and denoise in the latent space. For controllable synthesis, the conditional denoising autoencoder are implemented with text (Rombach et al., 2022; OpenAI, 2024) and various conditioning controls (Zhang et al., 2023; Sudhakar et al., 2024). We focus on video generation conditioned on the egocentric panoramic view of agent, as panorama (Li & Bansal, 2023; 2024) ensures coherence in the generated world, and the use of egocentric vision is de facto choice in many embodied tasks (Das et al., 2018; Sermanet et al., 2024; Song et al., 2025).

**Generative vision for embodied decision making.** Decision-making in the physical world (Das et al., 2018; Sermanet et al., 2024) is a fundamental AI challenge. LLMs provide linguistic reasoning that aids decision-making (Hao et al., 2023; Min et al., 2024) and vision-language planning (Cen et al., 2024). World models offer predictive representations of future states to inform decisions, though early attempts (Ha & Schmidhuber, 2018; LeCun, 2022) focus on simple game agents and often lack commonsense reasoning about the physical world. Generative vision (OpenAI, 2024; Kondratyuk et al., 2024) and in-context learning (Bai et al., 2024; Zhang et al., 2024) offer new avenues for using video generation to guide real-world decision-making (Yang et al., 2024c). Several works focus on specific application domains such as autonomous driving (Hu et al., 2023; Wang et al., 2023b; 2024c; Gao et al., 2024a;b) which limit their generality. Others like video in-context learning (Zhang et al., 2024) requires a known demonstration video, which is inefficient for decision-making. Action-conditioned video generation models (Du et al., 2024a; Yang et al., 2024b;c; Wang et al., 2024a; Bu et al., 2024; Souček et al., 2024; Du et al., 2024b) can directly synthesize visual plans for decision-making. These models, however, focus on state transition probabilities without explicitly modeling agent beliefs, which are crucial for reasoning about other objects/agents in partially observable environments.

## 3    GENERATIVE WORLD EXPLORATION

A machine explorer, such as a home robot, is designed to navigate within its environment and seek out previously unvisited locations. Integrating generative models, we present the concept of a **gen**erative world **ex**plorer (*GenEx*), enabling spatial exploration within an imaginative realm, akin to human mental exploration. We introduce the macro-design of *GenEx* in § 3.1, followed by the micro-design including input representation, diffuser backbone, and loss objective in § 3.2.

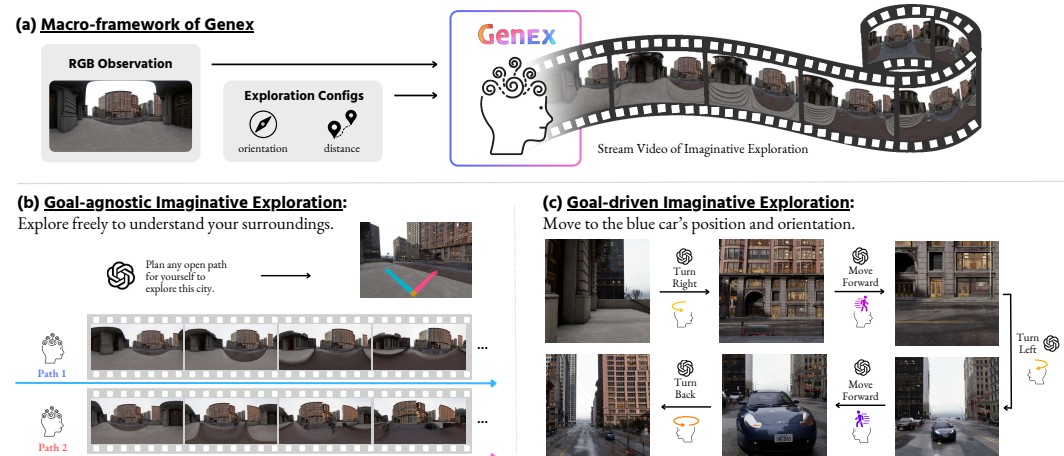

Figure 2: GenEx is able to explore an imaginative world by generating video sequence, given RGB observations, exploration direction, and distance (a). GenEx, grounded in physical environment, can perform GPT-assisted goal-agnostic imaginative exploration of the world (b) and goal-driven imaginative exploration (c).

### 3.1 MACRO-DESIGN OF *GenEx*

**(a) Overview**. As shown in Fig. 2, the GenEx framework enables agents to explore within an imaginative world by streaming video generation, based on current RGB observations and given exploration configurations. The RGB observation is represented as a panorama image sampled from any location in the world. A large multimodal model (LMM) serves as the pilot, or the decision maker, to set up exploration configurations, including any $360°$ navigation direction and distance. GenEx processes the input in two steps. First, it takes the exploration orientation to update the panorama forward view. Secondly, its built-in diffuser generates the forward navigation video. Both view update and diffuser are detailed in § 3.2.

*GenEx*, grounded in physical environment, can perform GPT-assisted goal-agnostic imaginative exploration and goal-driven imaginative exploration.

**(b) Goal-agnostic Imaginative Exploration**. *GenEx* can explore freely with an unlimited number of orientations, helping the agent to understand its surrounding environment, as shown in Fig. 2 (b).

**(c) Goal-driven Imaginative Exploration**. The agent receives a target instruction, such as, "Move to the blue car's position and orientation." GPT performs high-level planning based on the instruction and initial image, generating low-level exploration configurations in an iterative manner. GenEx then processes these configurations step-by-step, updating images progressively throughout the imaginative exploration as in Fig. 2 (c). This allows for greater control and targeted exploration.

### 3.2 MICRO-DESIGN OF *GenEx*

We detail our micro-design as follows. Our diffuser builds upon the standard stable video diffusion (SVD) (Blattmann et al., 2023a) backbone (a), but adapts the input from conventional images to panoramas (b). Furthermore, we propose spherical-consistent learning (c) to ensure coherence in imaginative exploration.

**(a) Diffuser backbone.** To support exploration illustrated in Fig. 2, we propose a video diffuser that can be seamlessly adapted to be a world explorer. Given an initial panorama image $x^0$ with camera position $p^0$, our objective is to generate a sequence of images $\{x^1, \ldots, x^n\}$ corresponding to a sequence of camera positions $\{p^1, \ldots, p^n\}$. The camera positions progress steadily forward, representing navigation in the world. Since the panorama image represents a 360-degree view, the generation should persist the information stored in previous frames to maintain world consistency throughout the sequence. Our model uses the pretrained SVD (Blattmann et al., 2023a). The Transformer UNet (Ronneberger et al., 2015; Chen et al., 2021) architecture is as described in Blattmann et al. (2023b), where temporal convolution and attention layers are inserted after every spatial convolution and attention layer. The pipeline of our model is shown in Fig. 3 (a). Given an image condition $c$ (encoded from image $x^0$ using CLIP image Transformer (Radford et al., 2021)), the video diffusion algorithms learn a network $\epsilon_\theta$ to predict the noise added to the noisy image latent $z_t$ with $\mathcal{L}_{\text{noise}} = \|\epsilon_\theta(z_t, c) - \epsilon_t\|^2$.

**(b) Input image representation.** Panorama images are well-suited for generative exploration as they captures all perspectives from an egocentric viewpoint into a 2D image. Essentially, it represents a spherical polar coordinate system $\mathcal{S}$ on a 2D grid in the Cartesian coordinate system $\mathcal{P}$, as shown in Fig. 3 (b). Panorama effectively stores every perspective of the world from a single location which preserves the global context during spatial navigation. This allows us to maintain consistency in world information from the conditional image, ensuring that the generated content aligns coherently with the surrounding environment. The panorama image also allows for *rotational transformations*, which facilitate *world navigation* by enabling us to rotate the image to face a different angle while preserving its original information. The rotation can be performed using Eq. 1:

$$\mathcal{T}(u, v, \Delta\phi, \Delta\theta) = f_{\mathcal{S}\to\mathcal{P}}\left(\mathcal{R}\left(f_{\mathcal{P}\to\mathcal{S}}(u, v), \Delta\phi, \Delta\theta\right)\right), \tag{1}$$

where $u$ and $v$ are positions on the 2D image plane, and $\phi$ and $\theta$ represent longitude and latitude in polar coordinates. The rotation function $\mathcal{R}$ applies a rotation to the spherical representation in any direction, simulating turning around during navigation. Additionally, a panorama image can be converted into a *cubemap* of six separate regular images, each representing a face of a cube (front, back, left, right, top, and bottom). This panorama-to-cube transformation enhances visual understanding by LMM agents. Full mathematical details of the equirectangular projection are in § A.1.

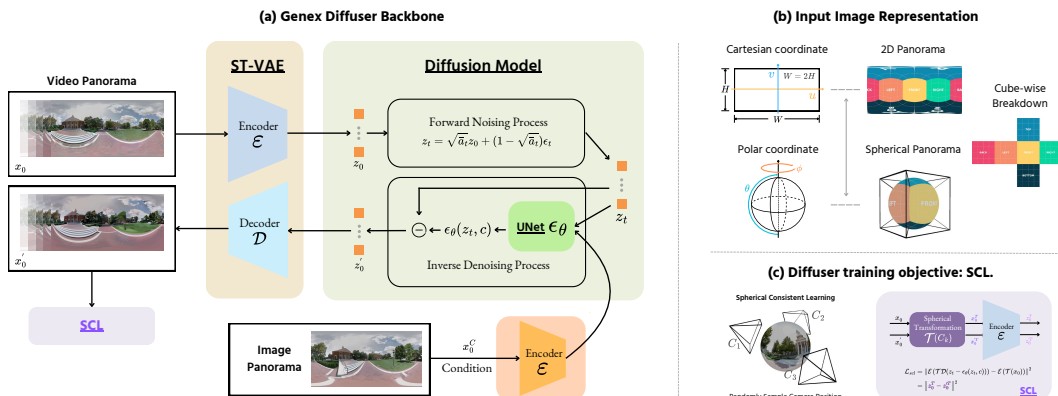

Figure 3: (a) Diffuser in *GenEx*, a spherical-consistent panoramic video generation model. During training, video $x_0$ is encoded into latent $z_0$ and noised to $z_t$. A conditioned UNet $\epsilon_\theta$ predicts and removes noise, resulting in $z_0'$ which is decoded to $x_0'$. The loss $\mathcal{L}_{scl}$ in (c) is combined with the original noise prediction loss. During inference, random noise is iteratively denoised to generate video $x_0'$ from an image panorama condition. (b) Left: Conversion between Polar and Cartesian coordinates. Right: Rotated spherical panorama can be converted to either 2D panorama or six-view images. (c) *Spherical-consistent learning*: we randomly sample camera orientation for edge consistency.

**(c) Diffuser training objective: spherical-consistent learning (SCL).** We aim to generate images where pixels are continuous in spherical space. However, direct training with results in severe edge inconsistency, as generated pixels at the far left and far right of the equirectangular image are not constraint to be continuous on the spherical space. To address this, we introduce spherical-consistent learning as an explicit regularization. After generating a panoramic video, we apply the spherical rotational transformation function, as shown in Eq. 1, to randomly rotates the camera to different positions on both the generated video and the ground truth, as illustrated in Fig. 3 (a). The denoised diffused video $x_t - \epsilon_\theta(x_t, c)$ and the ground-truth video $x_0$ are transformed and then passed into a pre-trained temporal VAE encoder $\mathcal{E}$, resulting in the latent over transformed diffused video $\mathcal{E}(\mathcal{T}(x_t - \epsilon_\theta(x_t, c)))$ and the latent over transformed ground-truth video $\mathcal{E}(\mathcal{T}(x_0))$. Each camera view is weighted equally in this process to ensure consistent representation across all perspectives. We train with the objective $\mathcal{L}_{scl}$ to minimize the mean square error over the latent spaces for maintaining uniformity and coherence in the 360-degree output. During training, the overall training objective is to minimize the loss:

$$\mathcal{L} = \lambda \underbrace{||\mathcal{E}(\mathcal{T}(\mathcal{D}(z_t - \epsilon_\theta(z_t, c)))) - \mathcal{E}(\mathcal{T}(x_0))||^2}_{\mathcal{L}_{scl}} + \underbrace{(1-\lambda)||\epsilon_\theta(z_t, c) - \epsilon_t||^2}_{\mathcal{L}_{\text{noise}}}, \quad (2)$$

where $\mathcal{D}$ is the temporal VAE (Kingma, 2013) decoder, $\lambda$ is a weighting constant, and $\mathcal{T}$ is the spherical rotation transformation shown in Eq. 1.

During inference, one can initialize $Z_{t_{\max}} \sim \mathcal{N}(0, \mathbf{I})$, iteratively sample $Z_{t-1} \sim p_\theta(Z_{t-1}|z_t, c)$ using the reparameterization trick, producing latent $z_0'$, which is decoded to panoramic video $x_0'$.

## 4  *GenEx*-BASED EMBODIED DECISION MAKING

### 4.1  IMAGINATION-DRIVEN BELIEF REVISION

Embodied agents operate under a POMDP framework (Puterman, 1994; Kaelbling et al., 1998). At each time step $t$, the agent's world state (which represents the complete environment at this specific moment), $s^t \in S$, and action $a^t \in A$ determine the next world state via the transition probability $T(s^{t+1}|s^t, a^t)$. The agent's given goal $g \in G$ (e.g., crossing the street) influences the reward $r^t = R(s^t, a^t, g)$, which drives the agent to achieve its objective. The agent receives an observation $o^t \in \Omega$ based on the observation model $O(o|s^t)$ and maintains a belief, represented by a distribution $b(s)$, which is the agent's internal estimate of the true state of the world. Its belief is updated with new observations, following the POMDP framework in Eq. 3:

$$b^{t+M}(s^{t+M}) = \prod_t \underbrace{O(o^{t+1}|s^{t+1}, a^t) \sum_{s^t} T(s^{t+1}|s^t, a^t)}_{\text{Physical Exploration}} b^t(s^t) \tag{3}$$

The decision $a^t$ made at any time $t$ becomes more informed as the agent gains a clearer understanding of its surroundings. By navigating through physical space, the agent gathers additional information about its environment (Fan et al., 2024), enabling more accurate assessments and better choices moving forward. However, physically traversing the space is inefficient, expensive, and even impossible in dangerous scenario. To streamline this process, we can use imagination as a medium for the agent to simulate outcomes without physically traversing. The key question becomes:

*How can an agent **revise** its belief through **imaginative exploration** for more **informed decisions**?*

**Imagination-driven belief revision**. We propose *imagination-driven belief revision* that uses imaginative exploration to enhance POMDP agents with a instant belief revision between time steps. In imagination, we freeze the time and create an imagined world, thus dropping the time variable $t$ and defining an *imagination space* with hat ^ on the variables. Here the agent can make a sequence of imaginative actions $\hat{\mathbf{a}} = \{\hat{a}_i \in \hat{A}\}$ over imagination time step $I = \{1, ...i, ..., n\}$. The agents can make sequential speculation on the unobserved world based on its initial belief and toward ultimate goal, it imagine novel observations in a previous unobserved world with $p_\theta(\hat{o}^{i+1}|\hat{o}^i, \hat{a}^i)$, where $\hat{o}^0 = o^0$ as initialization and $\theta$ is our parameterized video diffuser generating the imaginative observations. As a result, it can update its belief with Eq. 4:

$$\hat{b}^t(s^t) = \prod_i \underbrace{p_\theta(\hat{o}^{i+1}|\hat{o}^i, \hat{a}^i)}_{\text{Imaginative Exploration}} b^t(s^t) \tag{4}$$

Different from Eq. 3, we replace physical with imaginative exploration (Fig. 4). For an proper imagination, we should expect $b^{t+T}(s^{t+T}) \equiv \hat{b}^t(s^t)$, where the imaginative belief approximates the physical belief. As the sequence of imagination $I$ expands, more observations $o_i$ is produced, the agent's belief will be *approaching*, $b^*$, which is the belief the agent could obtain under a full observation.

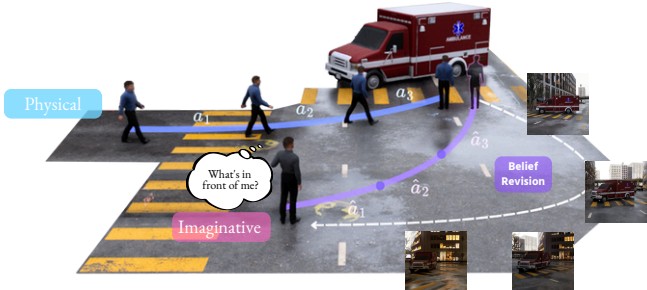

Figure 4: Imaginative exploration can achieve the same belief update as physical exploration.

The agent make actions based on its *belief* and *goal*, with policy model $\pi(a^t|b^t(s^t), g)$. Through the revised belief, the agent is capable of making more informed decision toward $a^*$ with a more refined belief toward $b^*$, with more information on the true state of its surrounding environment.

In our work, we apply *GenEx* for imaginative exploration and a LMM as the policy model $\pi$ and belief updater $b(s)$, mapping observation to belief, with examples in Fig. 5 and system pipeline in § A.5.3.

## 4.2 GENERALIZED TO MULTI-AGENT

Imagination-based POMDP can be generalized to the multi-agent scenario. The 1-st agent can imaginatively explore to the location of the $k$-th agent to predict the agent-$k$'s observation $\hat{o}_k$ and infer agent-$k$'s belief $\hat{b}_k$, following Eq. 4.

Thus, we can adjust agent-1's beliefs by aggregating the imagined belief counterpart for other $K-1$ agents.

$$a_1^t = \pi(\mathbf{b^K} = \{b_1, ...b_K\}, g) \tag{5}$$

When exploring another agent's thoughts, we can predict what that agent sees, understands, and might do next, which in turn helps us adjust our own actions with more complete information.

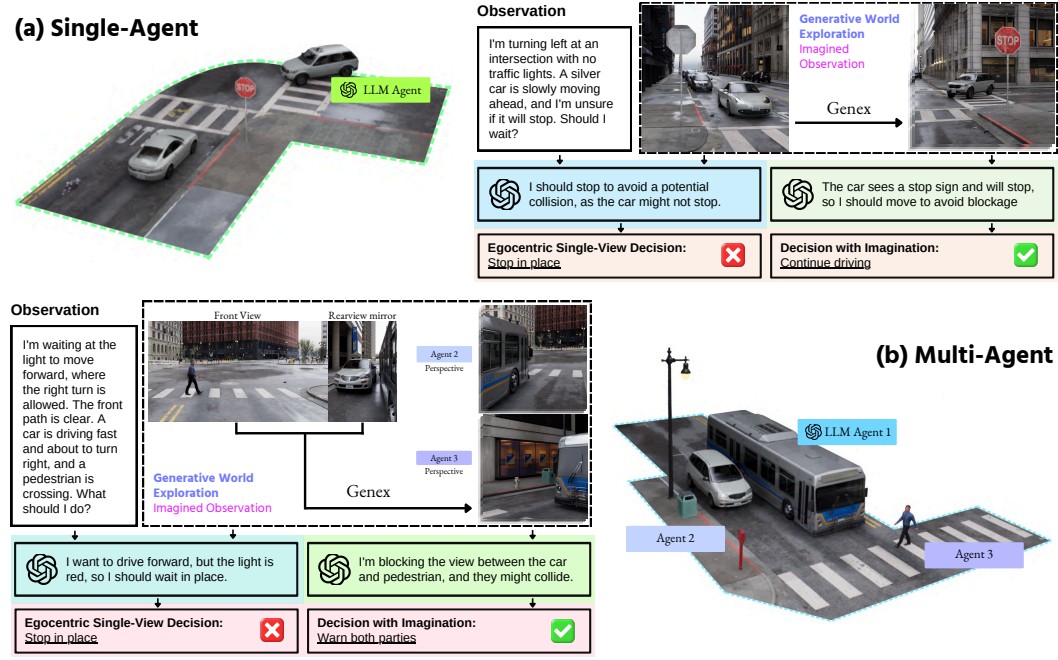

Figure 5: Single agent reasoning with imagination and multi-agent reasoning and planning with imagination. (a) The single agent can imagine previously unobserved views to better understand the environment. (b) In the multi-agent scenario, the agent infers the perspective of others to make decisions based on a more complete understanding of the situation. Input and generated images are panoramic; cubes are extracted for visualization.

We define embodied agents and introduce imagination-driven belief revision in § 4.1, followed by multi-agent decision making in § 4.2, and instantiation of embodied QA in § 4.3.

## 4.3 Instantiation in Embodied QA.

While the traditional EmbodiedQA benchmark (Das et al., 2018) features well-defined tasks such as navigation, they are not focus on how mental imagination help planning and the lack of multi-agent scenario limits further advancements (it doesn't satisfy condition (3)&(4) as follows). To the best of our knowledge, no existing benchmark can be used to evaluate our proposed solutions.

To bridge this gap, we aim to collect a new embodied QA benchmark satisfying four conditions: (1) The agent is planning with partial observation. (2) Questions can't be solved by linguistic common-sense alone; agents must physically navigate or mentally explore the environment to answer. (3) Humans can mentally simulate environments to comprehend and answer questions, but it's unclear if machines can do the same. (4) The benchmark can be extended to scenarios involving multi-agent decision-making. Accordingly, we propose a new dataset called GenEx-EQA in § 5.1.

## 5 Experiments

### 5.1 Dataset construction

**GenEx-DB.** We synthesize a large-scale dataset generated using Unity, Blender, and Unreal Engine. The full details are in § A.3. We create four distinct scenes, each representing a different visual style (*Realistic*, *Animated*, *Low-Texture*, and *Geometry*), shown in Fig. 6: We train a model with each dataset, and for the four resulting navigational video diffusers, we conduct cross-validation across all scenes to evaluate their generalization capabilities (detailed in § A.7).

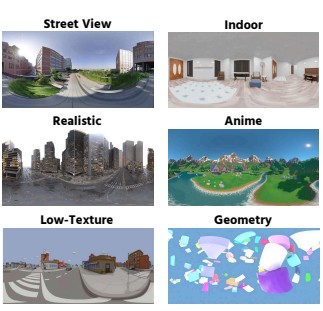

Figure 6: Examples for 6 different real and virtual scenes.

We collect an additional test set of panoramic images from *Google Maps Street View* (header "Street" in Table 2) and *Behavior Vision Suite* (Ge et al., 2024) (header "Indoor" in Table 2), which serves as a benchmark for real-world street and synthetic indoor exploration [1].

**GenEx-EQA**. Through the proposed *GenEx* model, the agents perform exploration autonomously, capable of tackling embodied tasks such as single agent scene understanding and multi-agent intersectional reasoning. Following the four conditions in § 4.3, we design over 200 scenarios in virtual physical engine to test various LMM agents embodied decision-making. We provide comprehensive details in § A.4.1. The dataset generally represent two scenarios:

- Single agent: the agent could infer the egocentric view from any location in its view. The agent could use *GenEx* to imagine the missing view (e.g. an ambulance blocked by trees *or* from the back of the stop sign). This extra information enables the agent to make more informed decisions.
- Multi-agent: the first agent can imaginatively explore the locations of other agents and use these imagined observations to update its beliefs.

## 5.2 EVALUATION ON GENERATION QUALITY

We adopt FVD (Unterthiner et al., 2019), SSIM (Wang et al., 2004), LPIPS (Zhang et al., 2018), and PSNR (Horé & Ziou, 2010) to evaluate video generation quality, with details in § A.5.

As a strong baseline, we develop a six-view navigator by training six separate diffusion model for each face of the cube, representing still a 360° view, independently (See Fig. 3 (b) six-view). The implementation detail is shown in § A.6. This baseline may align well with 2D diffusion models but stands in contrast to the panoramic approach, which is particularly effective at maintaining consistent environmental context. To enable a fair comparison with GenEx in video quality evaluation, the six-view

| Model | Input | FVD ↓ | MSE ↓ | LPIPS ↓ | PSNR ↑ | SSIM ↑ |
|---|---|---|---|---|---|---|
| → *direct test* | | | | | | |
| CogVideoX | six-view | 4451 | 0.30 | 0.94 | 8.89 | 0.07 |
| CogVideoX | panorama | 4307 | 0.32 | 0.94 | 8.69 | 0.07 |
| SVD | six-view | 5453 | 0.31 | 0.74 | 7.86 | 0.14 |
| SVD | panorama | 759.9 | 0.15 | 0.32 | 17.6 | 0.68 |
| → *tuned on **GenEx-DB*** | | | | | | |
| Baseline | six-view | 196.7 | 0.10 | 0.09 | 26.1 | 0.88 |
| GenEx w/o SCL | panorama | 81.9 | 0.05 | 0.05 | 29.4 | 0.91 |
| **GenEx** | panorama | **69.5** | **0.04** | **0.03** | **30.2** | **0.94** |

Table 1: Video generation quality of different diffusers.

baseline predictions are reprojected into panoramas. As a result, Table 1 shows that our method achieves high generation quality and surpass six-view baseline in all metrics.

## 5.3 EVALUATION ON IMAGINATIVE EXPLORATION QUALITY

Inspired by loop closure (Newman & Ho, 2005), we propose a new metric, **Imaginative Exploration Loop Consistency (IELC)**, to assess the coherence and fidelity of long horizontal imaginative exploration. *Definition*: For any randomly sampled path forming a closed loop within the scene, we calculate the latent MSE between the initial real image and the final generated image, both encoded by Inception-v4 (Szegedy et al., 2017). The final latent MSE is averaged over 1000 randomly sampled closed paths, with each loop differing in *number of rotations* and *total distance traveled* (refer to Fig. 7). We filter out paths blocked by obstacles.

In our results, we observe strong loop consistency across all exploration paths, shown in Fig. 8. Even in cases of long-range imaginative exploration (distance = 20m) and multiple consecutive videos, the latent MSE remained below 0.1, indicating minimal drift from the original frames. We attribute our method's strong performance to its preservation of spherical consistency in panoramas, ensuring that rotation does not degrade performance.

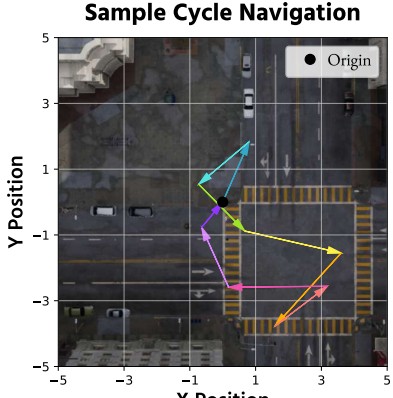

Figure 7: Example randomly sampled trajectory for loop consistency. It forms a *closed loop* within the scene, with 9 rotations and in 15 meters distance.

---

[1] For training, we exclude Google Maps Street View, for to its inconsistent image quality and unpredictable camera movement, and Behavior Vision Suite, for its restricted indoor navigation range.

We further conduct more analysis with three findings regarding the zero-shot generalizability to real-world, the correlation between generation and imaginative exploration, and the emerging 3D consistency below.

*Finding 1. The better the generation quality, the more consistent the imaginative exploration will be.* Fig. 9 shows a strong correlation between imaginative exploration loop consistency and generation FVD, validating our efforts to enhance the diffuser.

*Finding 2. GenEx, trained on synthetic data, demonstrates robust zero-shot generalizability to real-world scenarios.* Impressively, the model trained on UE5 and other synthetic data (Table 2), is generalized well (IELC $\leq$ 0.1) to indoor behavior vision suite and outdoor google map street view in real world, without requiring additional fine-tuning (See § A.7 for examples).

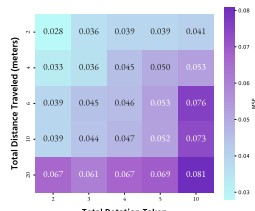 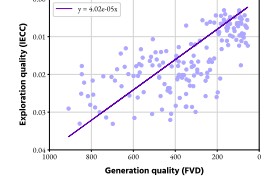

Figure 8: Imaginative Exploration Loop Consistency (IELC) varying distance and rotations.

Figure 9: Correlation between exploration quality (IELC) and generation quality (FVD).

| IELC ↓ | GenEx | | | | GenEx w/o SCL | Six-view |
|---|---|---|---|---|---|---|
| | Realistic | Anime | Low-Texture | Geometry | Realistic | Realistic |
| Street | **0.105** | 0.131 | 0.122 | 0.147 | 0.131 | 0.269 |
| Indoor | **0.092** | 0.168 | 0.103 | 0.117 | 0.120 | 0.233 |

Table 2: Zero-shot generalizability to real world. Rows are by zero-shot test scenes Columns are by models and training scenes.

*Finding 3. Generative world exploration empowers strong 3D understanding.* Our method enables the generation of multi-view videos of an object through imaginative exploration with a path circling around it. Table 3 not only report the common object-level foreground metric ($MSE_{obj.}$) but also highlight background evaluation ($MSE_{bg.}$). Our model demonstrates superior performance compared with the SoTA open-source models. Importantly, it maintains near-perfect background consistency and effectively simulates scene lighting, object orientation, and 3D relationships. Interestingly, we demonstrate that our model can reconstruct 3D worlds using additional plug-and-play models (Depth Anything (Yang et al., 2024a) & DUSt3R (Wang et al., 2024b)), as detailed in § A.8.

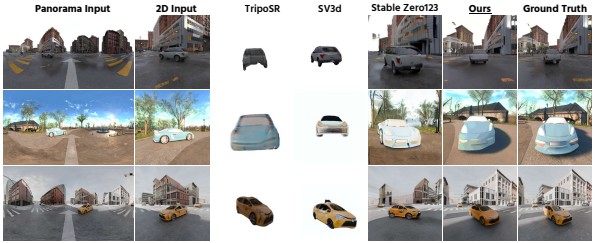

Figure 10: Comparison with state-of-the-art 3D reconstruction models for novel view synthesis. Through exploration, our model achieves higher quality in novel view synthesis for objects and improved consistency in background synthesis.

| Model | LPIPS↓ | PSNR↑ | SSIM↑ | $MSE_{obj.}$↓ | $MSE_{bg.}$↓ |
|---|---|---|---|---|---|
| TripoSR (Tochilkin et al., 2024) | 0.76 | 6.69 | 0.56 | 0.08 | - |
| SV3D (Voleti et al., 2024) | 0.75 | 6.63 | 0.53 | 0.08 | - |
| Stable Zero123 (StabilityAI, 2023) | 0.50 | 14.12 | 0.57 | 0.07 | 0.06 |
| **GenEx** | **0.15** | **28.57** | **0.82** | **0.02** | **0.00** |

Table 3: GenEx can synthesize novel views of distant objects (and background scene) with minimal difference from the ground truth, surpassing SoTA methods.

In summary, the robust zero-shot generalizability to real-world, the high correlation between generation and imaginative exploration, and the emerging 3D consistency pave the way for real-world embodied decision-making.

## 5.4 RESULTS ON EMBODIED QA

**Evaluation of embodied QA.** For embodied reasoning evaluation, we define three metrics:

- Decision Accuracy: this metric evaluates whether an agent's decision aligns with the optimal action a fully informed human would take. It measures the degree to which the chosen action successfully addresses the situation or problem.
- Gold Action Confidence: this refers to the agent's strength of belief to take the most appropriate action based on the available information and context. The confidence is calculated as the averaged normalized logit of the agent outputting the correct choice.
- Logic Accuracy: this metric tracks the correctness of the logical reasoning process that leads to a decision. We use LLM-as-a-judge (GPT-4o) to evaluate the agent's thinking process, with the

provided correct chain of thoughts. It highlights the sequence of steps, inferences, and reflections an agent makes while navigating toward a final action.

In Table 4, we evaluate our single-agent (§ 4.1) and multi-agent (§ 4.2) decision making algorithms. We use *Unimodal* refers to agents receiving only text context, while *Multimodal* reasoning demonstrate LLM decision when prompted along with a egocentric visual view. *GenEx* showcases the performance of models equipped as agents with a cognitive world model.

| Method | Decision Accuracy (%) | | Gold Action Confidence (%) | | Logic Accuracy (%) | |
|---|---|---|---|---|---|---|
| | Single-Agent | Multi-Agent | Single-Agent | Multi-Agent | Single-Agent | Multi-Agent |
| Random | 25.00 | 25.00 | 25.00 | 25.00 | - | - |
| Human Text-only | 44.82 | 21.21 | 52.19 | 11.56 | 46.82 | 13.50 |
| Human with Image | 91.50 | 55.24 | 80.22 | 58.67 | 70.93 | 46.49 |
| Human with **GenEx** | 94.00 | 77.41 | 90.77 | 71.54 | 86.19 | 72.73 |
| Unimodal Gemini-1.5 | 30.56 | 26.04 | 29.46 | 24.37 | 13.89 | 5.56 |
| Unimodal GPT-4o | 27.71 | 25.88 | 26.38 | 26.99 | 20.22 | 5.00 |
| Multimodal Gemini-1.5 | 46.73 | 11.54 | 36.70 | 15.35 | 0.0 | 0.0 |
| Multimodal GPT-4o | 46.10 | 21.88 | 44.10 | 21.16 | 12.51 | 6.25 |
| **GenEx (GPT4-o)** | **85.22** | **94.87** | **77.68** | **69.21** | **83.88** | **72.11** |

Table 4: Embodied QA evaluation across different scenarios. For unimodal input, agent is prompted with only text context, and for multimodal input, agent is given its egocentric image view. In all settings, we prompted the agent to generate in Chain-of-Thoughts to image other agent's belief.

*Vision without imagination can be misleading for GPTs.* In some cases, the unimodal's response (processing only the environment's text description) surpasses the multimodal counterparts (which includes both text and egocentric visual input). This suggests that vision without imagination can be misleading. When an LLM agent converts its view into a text description and relies solely on language-based commonsense reasoning, it tends to make incorrect inferences due to the lack of spatial context. This highlights the importance of integrating imagination with visual data to enhance the accuracy and reliability of the agent's decision-making processes.

*GenEx has potential to enhance cognitive abilities for humans.* Human performance results reveal several key insights. First, individuals using both visual and textual information achieve significantly higher decision accuracy compared to those relying solely on text. This indicates that multimodal inputs enhance reasoning. Secondly, when provided with imagined videos generated by GenEx, humans make even more accurate and informed decisions than in the conventional image-only setting, especially in multi-agent scenarios that require advanced spatial reasoning. These findings demonstrate GenEx's potential to enhance cognitive abilities for effective social collaboration and situational awareness.

# 6 CONCLUSION

We introduced the Generative World Explorer (*GenEx*), a novel video generation model that enables embodied agents to imaginatively explore large-scale 3D environments and update their beliefs without physical movement. By employing spherical-consistent learning, *GenEx* generates high-quality and coherent videos during extended exploration. Additionally, we present one of the first methods to integrate generative video into the partially observable decision-making process through imagination-driven belief revision. Our experiments show that these imagined observations significantly enhance decision-making, allowing agents to create more informed and effective plans. Furthermore, *GenEx*'s framework supports multi-agent interactions, paving the way for more advanced and cooperative AI systems. This work marks a significant advancement toward achieving human-like intelligence in embodied AI.

## 7 ACKNOWLEDGEMENT

This work is partially supported by ONR with award N000142412696 to AY and a Siebel Scholarship to JC. We are grateful to the anonymous reviewers for their constructive feedback and to the many wonderful researchers at JHU who contributed after our ICLR submission.

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

# A APPENDIX

## A.1 PRELIMINARY: EQUIRECTANGULAR PANORAMA IMAGES

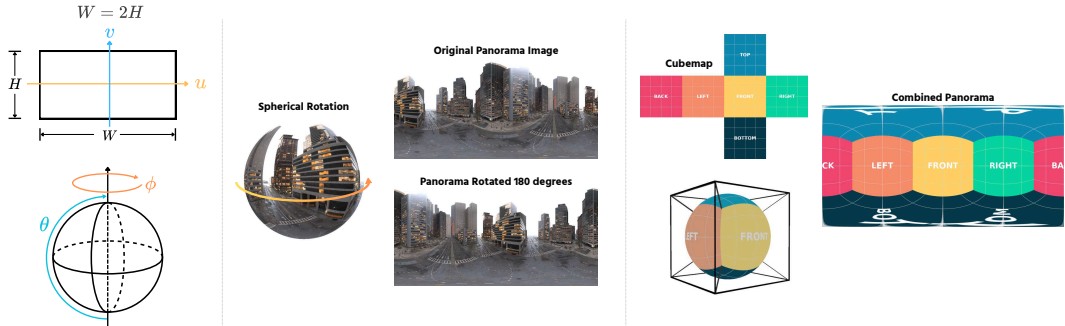

Figure 11: Left: Pixel Grid coordinate and Spherical Polar coordinate systems; Middle: rotation in Spherical coordinates corresponds to rotation in 2D image; Right: expansion from panorama to cubemap or composition in reverse.

### A.1.1 COORDINATE SYSTEMS

An *Equirectangular Panorama Image* captures all perspectives from an egocentric viewpoint into a 2D image. Essentially, it represents a spherical coordinate system on a 2D grid.

**Definition D.1** (Spherical polar coordinate system). $\mathcal{S}$: Taking the origin as the central point, a point in this system is represented by coordinates $(\phi, \theta, r) \in \mathcal{S}$, where $\phi$ denotes the longitude, $\theta$ the latitude, and $r$ the radial distance from the origin. The ranges for these coordinates are $\phi \in [-\pi, \pi)$, $\theta \in [-\pi/2, \pi/2]$, and $r > 0$.

**Definition D.2** (Cartesian coordinate system for panoramic image). $\mathcal{P}$: In this system, a pixel is identified by the coordinates $(u, v) \in \mathcal{P}$, where $u$ and $v$ correspond to the column and row positions on the 2D panoramic image plane, respectively. Here, $u$ ranges from 0 to $W - 1$ and $v$ ranges from 0 to $H - 1$.

**Definition D.3** (Sphere-to-Cartesian Coordinate Transformation). The transformation between the spherical polar coordinates and the panoramic pixel grid coordinates can be defined by the following functions:

$$f_{\mathcal{S} \to \mathcal{P}}(\phi, \theta) = \left( \frac{W}{2\pi}(\phi + \pi), \frac{H}{\pi}\left( \frac{\pi}{2} - \theta \right) \right) \tag{6}$$

$$f_{\mathcal{P} \to \mathcal{S}}(u, v) = \left( \frac{2\pi u}{W} - \pi, \frac{\pi}{2} - \frac{\pi v}{H} \right) \tag{7}$$

Here, the function $f_{\mathcal{S} \to \mathcal{P}}$ maps the spherical coordinates $(\phi, \theta)$ to the pixel coordinates $(u, v)$, and the inverse function $f_{\mathcal{P} \to \mathcal{S}}$ maps the pixel coordinates $(u, v)$ back to the spherical coordinates $(\phi, \theta)$. This transformation ensures that the entire spherical surface is represented on the 2D panoramic image.

Panorama effectively stores every perspective of the world from a single location. In our work, due to the nature of panoramic images, we are able to preserve the global context during spatial navigation. This allows us to maintain consistency in world information from the conditional image, ensuring that the generated content aligns coherently with the surrounding environment.

### A.1.2 PANORAMA IMAGE TRANSFORMATIONS

The spherical format allows various image processing tasks. For example, the image can be rotated by an arbitrary angle without any loss of information due to the spherical representation. Additionally, it can be broken down into cubemaps for 2D visualization, as shown in Fig. 11.

**Definition D.4** (Rotation Transformation in Spherical Polar Coordinate System)**.** Since a panorama image is in a spherical format, we can rotate the image to face a different angle while preserving the original image quality. The rotation can be performed using the following formula:

$$\mathcal{T}(u, v, \Delta\phi, \Delta\theta) = f_{\mathcal{S}\to\mathcal{P}}\left(\mathcal{R}\left(f_{\mathcal{P}\to\mathcal{S}}(u, v), \Delta\phi, \Delta\theta\right)\right) \tag{8}$$

Where the rotation function $\mathcal{R}$ is defined as:

$$\mathcal{R}(\phi, \theta, \Delta\phi, \Delta\theta) = (\phi + \Delta\phi \,(\mathrm{mod}\, 2\pi), \theta + \Delta\theta \,(\mathrm{mod}\, \pi)) \tag{9}$$

If there is no explicit input, both $\Delta\phi$ and $\Delta\theta$ can be set to 0.

**Panorama to cubes** A panorama image can be broken down into six separate images, each corresponding to a face of a cube: front, back, left, right, top, and bottom, as shown in Fig. 11. This conversion allows the panorama to be viewed as six conventional 2D images.

## A.2 Hyperparameters and Efficiency of GenEx-Diffuser

We provide the training hyperparameters for GenEx diffuser in Table 5 and computation resource used for training in Table 6.

| Hyperparameters | Value |
|---|---|
| learning rate | 1e-5 |
| lr scheduler | Cosine |
| output height | 576 |
| output width | 1024 |
| mixed precision | fp16 |
| training frame | 25 |
| lr warmup steps | 500 |

Table 5: GenEx-Diffuser Training configuration.

| Setting | Value |
|---|---|
| Total GPU Usage | 384 A100 hours |
| GPU Configuration | 2 A100 per batch, Model Parallelism |
| Training Time | 0.12 minutes per step |
| Inference Time | 0.031 minutes per frame |

Table 6: GenEx-Diffuser Training and Inference Time.

## A.3 GenEx-DB

For dataset creation, we use scenes in four different styles to examine how different visual representation affect final model performance.

- Realistic: Using the Sample City from Unreal Engine 5, designed to evaluate the model's ability to handle photorealistic environments.
- Animated: Created to test the model's performance in stylized, animated settings.
- Low-Texture: Used to assess how well the model adapts to environments with minimal texture details, focusing on whether the model can learn relying only on architectures.
- Geometry: Composed solely of simple geometric shapes (cubes and cylinders), designed to determine if the model can learn panoramic movement from basic forms.

In an chosen 3D environment, we sample a random position and random rotation. We sample a path moving straight forward for 20 meters where there is no collision to any objects and render a video moving in this path with constant velocity for 50 frames. During training, we randomly sample a from $\mathrm{frame}_1$ to $\mathrm{frame}_2 5$ as the conditional image with ground truth the navigation in the next 25 frames. Image example is provided in Fig. 12.

We report dataset statistics in Table 7.

## A.4 GenEx-EQA

### A.4.1 Dataset details

Generally, the GenEx-EQA could be divided into two categories, Single-Agent and Multi-Agents. In single-agent scenario, the agent should be able to make the appropriate decision with only its

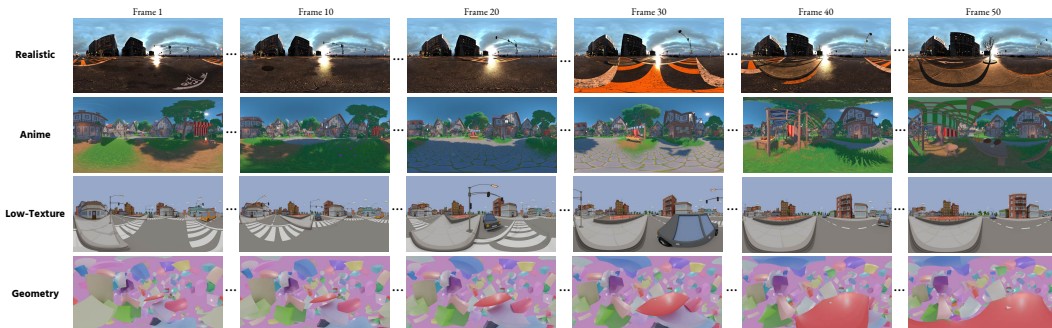

Figure 12: Dataset examples are four distinct scenes. Each sampled video consist of 50 frames. At each step, 25 frames are chosen for training.

| Statistics | Value |
|---|---|
| Engine (Environment) | UE5 (City Sample), Unity (Low-texture City, Animate), Blender (Geometry) |
| # scenes | 40000 + |
| # frames | 2,000,000 + |
| # traversal distance (m) | 400,000 + |
| # total time (s) | 285,000 + |
| # navigation direction | +inf |

Table 7: The data statistics for GenEx-DB.

current observation (it does mean there is only one agent exist in the scene). In multi-agent scenario, to fully understand the environment state, the agent need to understand what other agent's belief. For each test case, we repeat the scene in a low-texture virtual environment to observe the different in behavior from observation realistic level.

We provide examples on the constructed GenEx-EQA dataset in Fig. 13.

For each scenario, we include a control set. For example, if the exploration would end up with an ambulance driving toward the agent, there also exist an setting where the ambulance is driving away from the agent.

We report the statistics of GenEx-EQA dataset in Table 8.

| Statistics | Value |
|---|---|
| Engine | UE5, Blender |
| Environment | City Sample, Low-texture City |
| # scenes | 200 + |
| # agents | 500 + |
| # average agent per scene | 2.7 |
| # text context | 800 + |
| # actions | 200 + |
| # navigation direction | +inf |

Table 8: The data statistics of GenEx-EQA benchmark.

## A.5 QUANTITATIVE ANALYSIS IMPLEMENTATION

For all tested videos, FVD, LPIPS, PSNR , SSIM is calculated by resizing each image to $1024 \times 576$ pixels and comparing them with the ground truth videos at the same dimensions.

For latent MSE of images, each image is resized to $500 \times 500$ pixels and processed through the Inception v4 model Szegedy et al. (2017) to compute the latent MSE. When comparing IELC, we compare the latent MSE between beginning and ending frames.

## Single-Agent

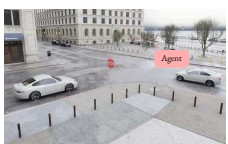

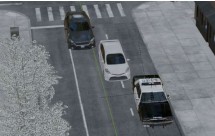

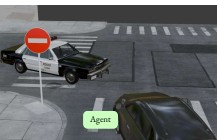

**Scene:** I arrive at an intersection and want to turn left. The front path is clear, there is no car. … I see another car at the intersection, on the left view moving slowly.

**Question:** What should I make the turn?

**Choices:**

(A) Stop in place to wait for the car to make the turn first.

(B) Honk to warn other cars to avoid collision.

(C) Pull over and wait for traffic to clear.

(D) Carefully continue the turn to avoid traffic congestion.

**Scene:** I arrive at an intersection and want to drive forward. … I see the car opposite to myself suddenly stop. Also, I hear what seems to be an alarm, possibly from an emergency vehicle."

**Question:** What should I do?

**Choices:**

(A) Change lanes to bypass the car carefully.

(B) Stop passing the intersection and move a little bit left to clear the way.

(C) Stop in place to observe the environment.

(D) Continue to proceed through the intersection since the traffic light is green.

**Scene:** I am driving down a street. Ahead, there is a car stopped in my lane. I can't see what is in front of this car because it is blocking my view. The traffic is light, …

**Question:** How should I proceed?

**Choices:**

(A) Change lanes to pass the stopped car quickly, since there is no visible obstruction.

(B) Honk to signal the stopped car to move.

(C) Slow down and keep to my lane, proceeding with caution.

(D) Wait for the car ahead to start moving.

**Scene:** I am approaching an intersection with a "Do Not Enter" sign. … Ahead, there is a police car in view, but it is unclear whether the police car is waiting or needs to move.

**Question:** How should I respond to this situation?

**Choices:**

(A) Wait at the intersection for the police car to move first.

(B) Change lanes to pass through.

(C) Honk to signal the police car to move.

(D) Slow down and proceed cautiously, assuming the police car will stay in place.

## Multi-Agent

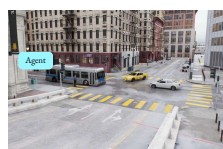

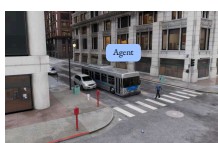

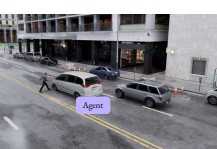

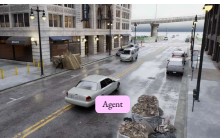

**Scene:** I arrive at an intersection to proceed forward. The intersection does not have a traffic light and is busy. There is a pedestrian on my right side crossing the road slowly.

**Question:** What should I do now?

**Choices:**

(A) Drive forward as normal.

(B) Block the pedestrian for a few seconds to avoid hitting by other cars.

(C) Accelerate to avoid collision with other cars.

(D) Pull over and wait for traffic to clear.

**Scene:** I'm at an intersection with a red light, where right turns are allowed. … A fast car is approaching to turn right, and a pedestrian is crossing in front of me.

**Question:** What do I need to do?

**Choices:**

(A) Signal the car to stop for the pedestrian.

(B) Stay in place and wait for the green light.

(C) Honk to alert the pedestrian of the approaching car.

(D) Proceed cautiously while monitoring both the car and pedestrian.

**Scene:** I'm driving on a street. The front path is clear. … I see a car in my back try to bypass me. There is also a pedestrian crossing the street on my left side.

**Question:** What would I do?

**Choices:**

(A) Move a little bit to the left to allow the other car to pass.

(B) Continue drive forward fast.

(C) Slow down to avoid the car bypass now to protect the pedestrian.

(D) Suddenly stop in place to block the back car.

**Scene:** I'm driving on the right lane on a street. On the other lane, there is a car approaching fast. … I can also see a pedestrian on the left side trying to cross the street.

**Question:** What to do now?

**Choices:**

(A) Continue forward as the path is clear.

(B) Honk to signal the front car to avoid collision with me.

(C) Pull over to the right.

(D) Warn both pedestrain and the car for a potentail collision.

Figure 13: Example GenEx-EQA questions. We generally divide the questions into two categories. (1) single-agent is testing the ability a agent to make optimal decision independent of social interaction. For example, in the first scene, decision agent need to infer what can the other car see, but it does not need to infer the belief that agent hold. (2) Multi-agent is testing the ability of agents to measure other agents' belief and their potential interaction. For example, in the first scene in the second row, the agent need to infer the pedestrian's belief in its surrounding and also the other car's belief.

### A.5.1 LMM PROMPT FOR WORLD EXPLORATION

We prompted LMM to navigate throughout the scene. The format is provided in Fig. 14. To handle difference in distance traveling, we use different number of frames from generation. For example, if the diffusion model generate 25 frames at once and one frame means traveling 0.4 meters, travel 4 meter would mean take the first 10 frames.

---

**LMM Navigation Prompt**

Scenario Context:
You are presented with a 360-degree equirectangular panorama, a type of image that captures the entire spherical view of an environment and flattens it into a rectangular format. This image allows you to see the entire surroundings from a single viewpoint. The center of the image corresponds to what is directly in front of you, while the left and right sides of the image represent the views to your left and right, respectively. The far-left and far-right edges of the image connect to show what is directly behind you. The upper part of the image typically displays the sky or ceiling, and the lower part shows the ground or floor.

Requirement:
Navigate through the urban street scene based on provided instructions. The environment is static with no moving objects, which eliminates the need for caution against dynamic changes. Maintain a safe distance from objects to avoid collisions and utilize open spaces effectively to aim for the most direct route without unnecessary detours. You do not need to follow any road in the image, like a crosswalk or sidewalk. You can move in any direction.

Navigation Process:
This task is part of a multi-step navigation process. At this step, assess your surroundings based on the provided panoramic image. Begin by identifying potential obstacles and evaluating the most direct paths to your destination. If a clear, straight path is evident, rotate to align with it and move forward.
Action Choices: (1) Turn left, (2) Turn right, (3) Move forward (4) Stop

Image-Specific Instruction:
{Instruction}

Analysis Request:
1. Scene Analysis: Start with a thorough examination of the scene. Describe what is visible in the left, front, right, and back views.
2. Target Identification: Identify where the target location is situated, considering the layout and nearby objects.
3. Path Planning: Plan an efficient and clear path. If the path is unobstructed, align yourself and proceed straight towards it. Ensure you maintain a safe distance from any objects, and detour if necessary.
4. Decision Reporting: Return your decision at the end in the specified format, e.g., [[move forward 1 meter]]. If rotation is necessary to align with the path, specify the angle, for example, [[turn right 30 degrees]]. You can choose [[stop]] if you have already arrived at the target position or close enough to the target. Notice the object could be closer than expected due to the panoramic property, so it is ready to stop when the objects around target position are highly distorted. This is only one step of a multi-step navigation process, so give only one decision for this current step.

Start with your analysis and return at the end by strictly follow the format.

Figure 14: Navigation prompt template.

### A.5.2 EMBODIED DECISION MAKING USING LMM

We provide LMMs with context using the prompt format illustrated in Fig. 15. In multimodal scenarios, we also include the egocentric (first-person) view, presented as six separate images, in addition to the unimodal cases.

### A.5.3 SYSTEM PIPELINE OF EQA DECISION-MAKING

We shows a general imagination-enhaced LMM POMDP system pipeline shown in in Fig. 16.

**LLM Prompt**

Below are the views provided to me from my surroundings.
<Image 1> This is the my __ view .
<Image 2> This is the my __ view .
...

I am currently operating a vehicle. Based on this information, and considering the given context, please recommend the most appropriate course of action for me to take.

Additionally, evaluate the perspectives and likely reactions of other drivers, pedestrians, or obstacles in the environment. How might their actions or behaviors influence your recommendation? Your guidance should be updated based on the evolving situation, ensuring that your final recommendation reflects a thorough assessment of the surroundings.

SCENE:
{scene text description}

Question:
{question}

OPTIONS:
{choices}

Start by providing your reasoning for each possible action, considering both my immediate environment and the actions of others. Conclude with a probability-based recommendation. The total probability across all options must equal 100%. Format your response strictly as: [A: XX%], [B: XX%], [C: XX%], [D: XX%], where "XX" represents the percentage probability assigned to each option.

* Image is provided only in Multimodal Setting

Figure 15: Embodied QA prompt template

Figure 16: EQA answer pipeline. It follows Imagination-enhanced POMDP, updating its belief with imagination for more informed decision.

### A.5.4 EVALUATION METRIC

**Machine Evaluation**. We provide the three embodied decision metrics to evaluate the benchmarked agents.

1. **Decision Accuracy and Confidence**. Since we prompted the LLMs to generate in a given format, we directly parse the accuracy and confidence. In case the LLM failed to follow the format, we filter and remove the cases.

2. **Decision Confidence**. This represents the agent's confidence in selecting the most appropriate action based on the available information and context. Confidence is calculated by averaging the normalized logits corresponding to the agent's correct choice.

3. **Chain-of-Thoughts Accuracy** This metric evaluates the accuracy of the agent's logical reasoning that leads to a decision. We employ GPT-4o as a judge to assess the agent's thought process against the correct chain of thoughts. It highlights the sequence of steps, inferences, and reflections the agent uses to reach a final action. The prompts to GPT-4o are provided in Fig. 17.

---

**LLM as a judge Prompt**

Please assess the logical correctness of the following response generated by an LLM.

Your task is strictly to evaluate whether the logic in the LLM's response aligns with the provided correct reasoning. Avoid introducing any personal prior beliefs or interpretations. Your sole responsibility is to verify if the logic in the reference is accurately reflected in the LLM's output.

After completing the assessment, return either [[YES]] or [[NO]], strictly adhering to this format.

LLM Response: {text}

Correct Logic: {logic}

---

Figure 17: The prompt template to GPT4o-as-a-judge.

**Human Evaluation**. We present the same prompt to all human evaluators. In unimodal scenarios (both realistic and stylized) we reuse the same results due to the absence of randomness, similar to fixed temperatures in LLMs. Evaluators are guided through three strict steps to prevent information leakage: (1) text description only, (2) egocentric view, and (3) pre-navigated GenEx generation. This sequential approach ensures consistency and maintains the integrity of the evaluation process.

### A.6 COMPARED METHOD: SIX-VIEW EXPLORATION

We use the same training configuration and dataset as in the original approach, but instead of working directly with panorama images, as in Fig. 18, we break the equirectangular image down into six faces of a cube. Each face corresponds to a specific direction: front, left, right, back, top, and bottom, as in Fig. 11, which obtain navigation process by focusing on discrete sections of the scene.

- Front view always moves forward.
- Left view moves to the right.
- Right view moves to the left.
- Back view moves backward.
- Top view remains stationary for upward and moves forward.
- Bottom view remains stationary for downward and moves forward.

Although each face provides a clear perspective, the transitions between faces introduce inconsistencies as information in cube faces are not shared. However, panorama navigation can preserve a general world context.

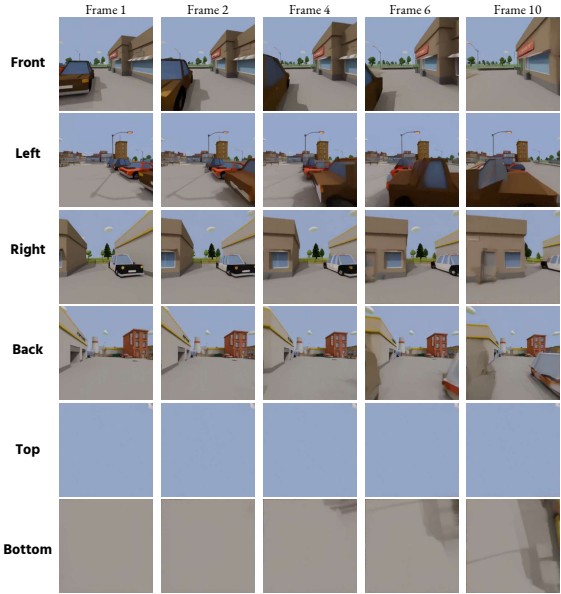

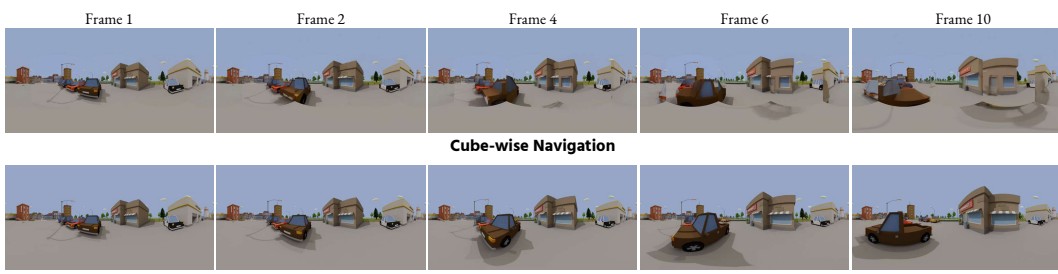

Figure 18: In six-view exploration baseline, we train 6 separate diffuser representing each cube face. Although each individual face remains acceptable quality, the world context is not preserved as in panoramic world exploration.

## A.7 DETAILS OF CROSS-SCENE GENERATION

The model shows a strong cross-scene generalization ability. From the loop consistency results in Table 2, the panorama generation works well even for scenes deviates largely from its training set.

**Dataset** For each model trained by the dataset described in § 5.1, we evaluate its cross-scene generation quality.

**Metric** We evaluate loop consistency for different scenes when trained on different model and report in Table 9.

| Loop Consistency | Street | Indoor | Realistic | Anime | Texture | Geometry |
|---|---|---|---|---|---|---|
| **Realistic** | 0.1051 | 0.0917 | 0.0687 | 0.1248 | 0.1332 | 0.2047 |
| **Anime** | 0.1044 | 0.1679 | 0.1171 | 0.0571 | 0.1347 | 0.2890 |
| **Low-Texture** | 0.1215 | 0.1032 | 0.1104 | 0.1624 | 0.0508 | 0.0800 |
| **Geometry** | 0.1471 | 0.0782 | 0.1230 | 0.1746 | 0.0685 | 0.0434 |

Table 9: Cross-Scene Loop Consistency (§ 5.2) Latent MSE by training scene and test scene. Columns are by test scenes and rows are by training scenes.

**Image Example** We demonstrate some example of cross-scene generation. For example, When training using the Anime dataset, the model can generalize to generate novel view of a car in the Low-Texture dataset, although nothing similar exist in its training set. More image examples are provided in Fig. 19.

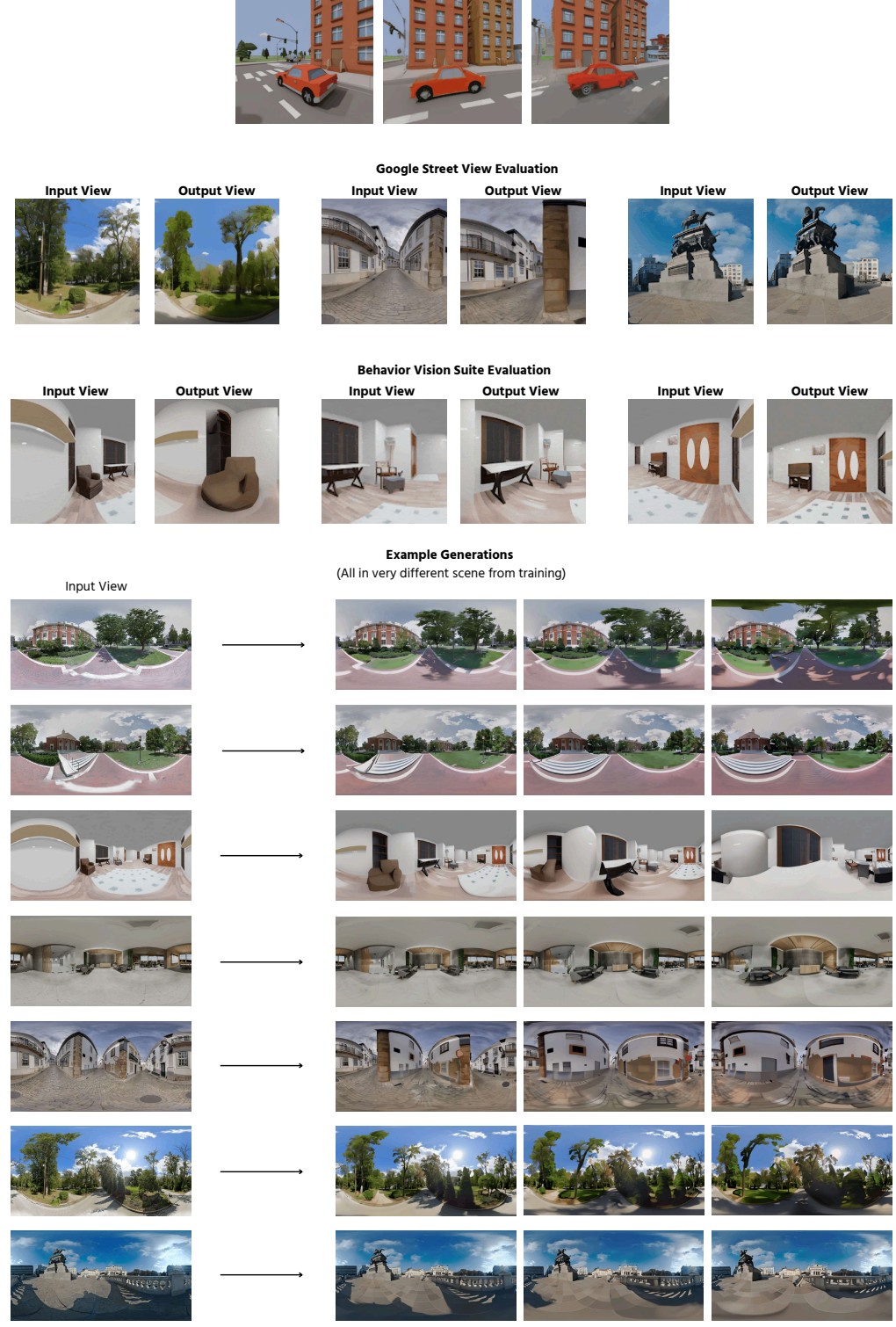

Figure 19: Cross-scene generation examples. Neither google street view or indoor scene is used for training (Inputs and outputs are panorama images. We extract cubes for visualization).

## A.8 EXTENSION TO 3D REPRESENTATION OF WORLD

**3D Egocentric World.** We are able to reconstruct a egocentric 3D point cloud combining single panorama image with the external tool of Depth-Anything-v2 (Yang et al., 2024a). Examples are shown in Fig. 20. For each point on the image, we directly map it to a 3D location using depth.

Given a pixel $(u, v)$ with image dimensions $W$ (width) and $H$ (height), each point $(X, Y, Z)$ represents a point in the 3D point cloud.:

**Compute angles:**     **Calculate 3D coordinates:**

$$\theta = \frac{2\pi u}{W} - \pi \qquad X = D \cdot \cos(\phi) \cdot \cos(\theta)$$
$$\phi = \pi\left(1 - \frac{v}{H}\right) - \frac{\pi}{2} \qquad Y = D \cdot \sin(\phi)$$
$$Z = D \cdot \cos(\phi) \cdot \sin(\theta)$$

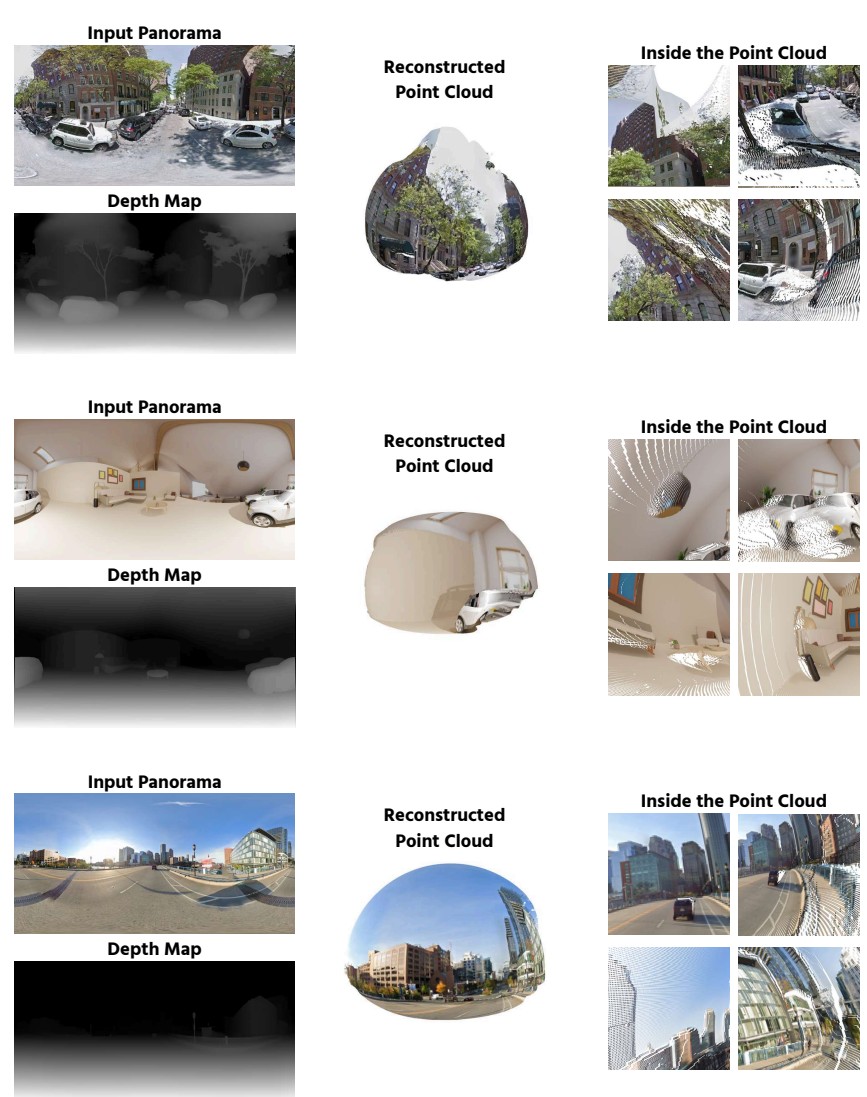

Figure 20: Egocentric 3D reconstruction with depth map and point cloud using monocular depth estimation tools (Yang et al., 2024a).

**3D Exocentric world.** To construct a exocentric 3D reconstruction from multi-view image. For any given panorama image, we could sample random forward moving direction to generate multiple

panorama image. Breaking down into cubes, panorama images become usable 2d images to be passed in reconstruction model like DUSt3R (Wang et al., 2024b). Examples are shown in Fig. 21.

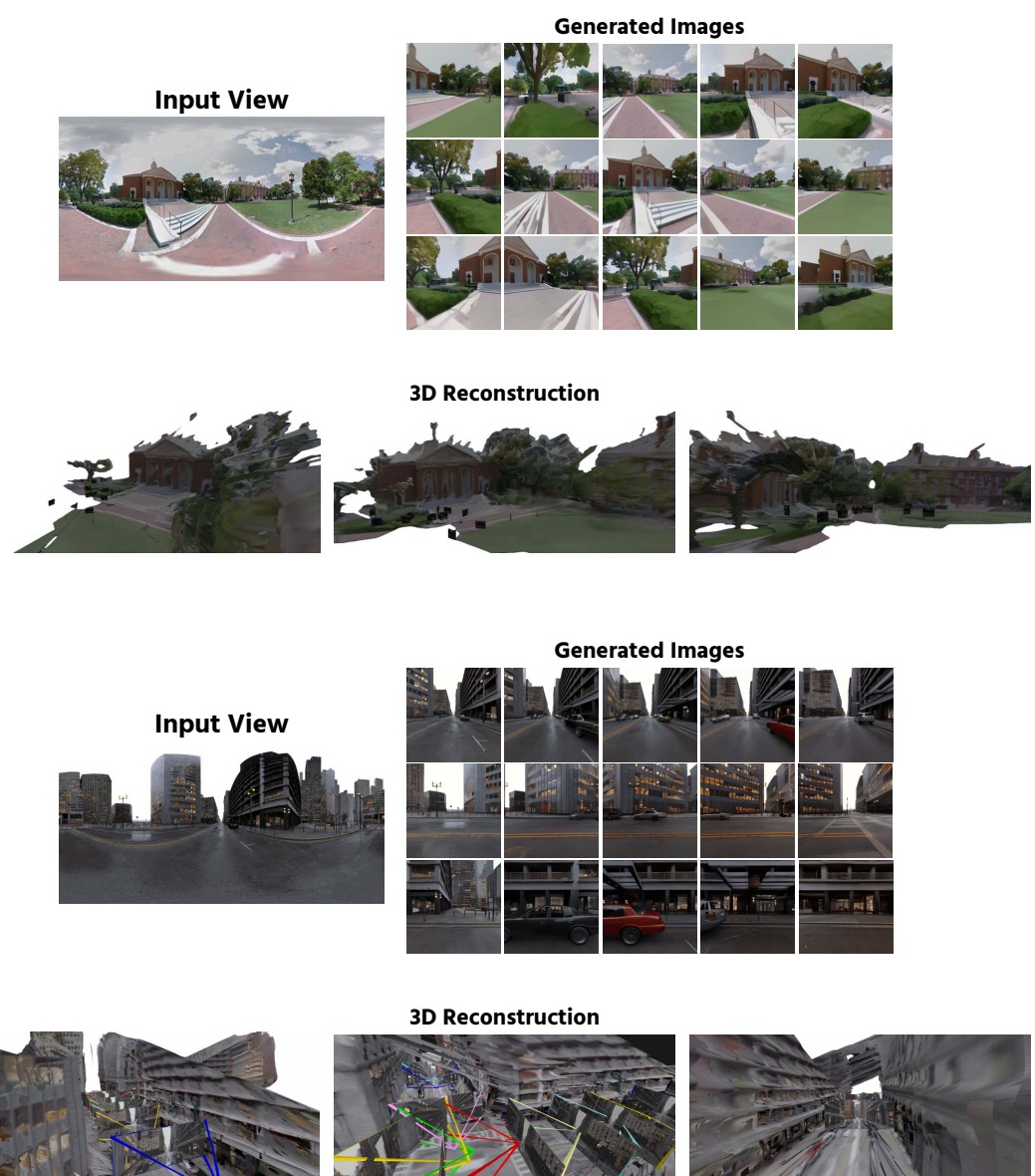

Figure 21: Exocentric 3D reconstruction with DUSt3R (Wang et al., 2024b).

## A.9 BIRD'S-EYE VIEW GENERATION

Our method can generate bird's-eye view (BEV) maps from a single panoramic image by leveraging exploration along the *z-axis*. By adjusting the exploration pipeline to navigate upwards, we can extract a top-down view directly. As shown in Fig. 22, examples of the generated BEV maps illustrate the effectiveness of our method in capturing a comprehensive top-down perspective from a single panoramic image. This capability enables the agent to imagine a third-person perspective through BEV maps, supporting more informed and objective decision-making.

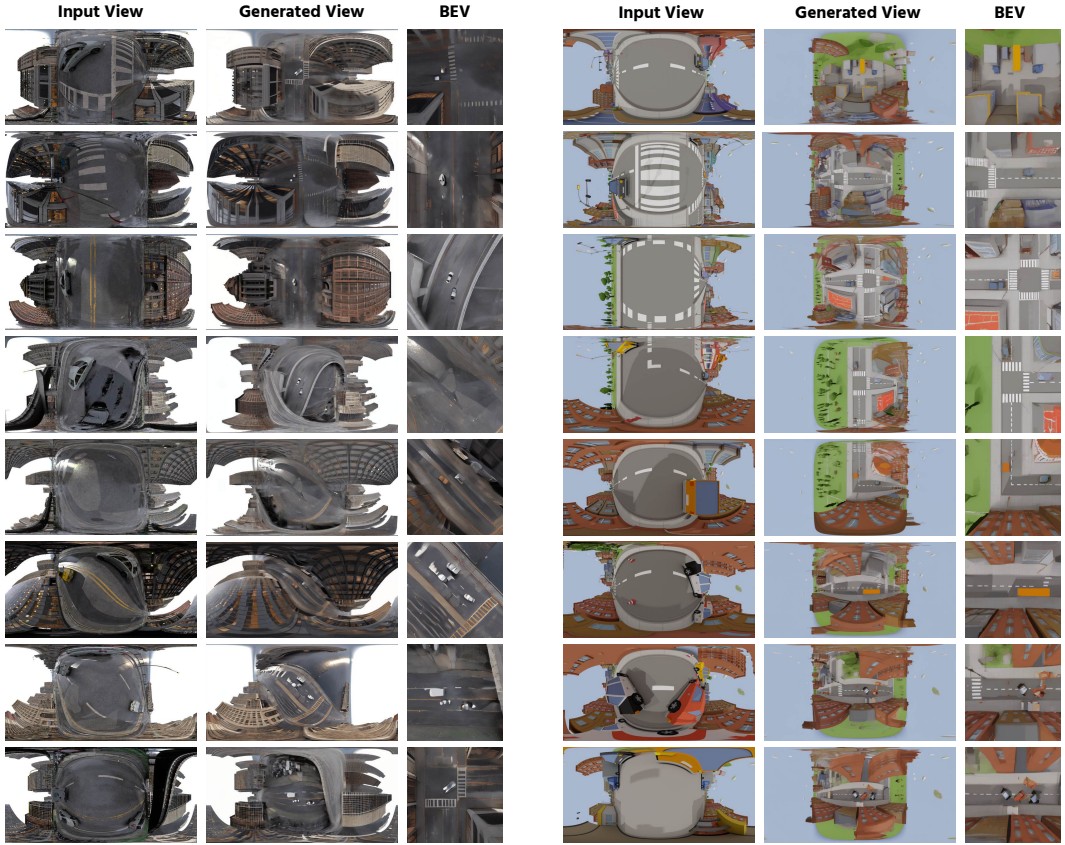

Figure 22: By exploration in *z-axis*, we are able to generate the 2D bird-eye view of the current scene.

