# OpenReview forum: "GenEx: Generating an Explorable World"
_ICLR.cc/2025/Conference — ICLR 2025 Poster_

### Official Review · Reviewer_GZnb · 2024-11-01

**Soundness:** 2
**Presentation:** 2
**Contribution:** 4
**Rating:** 6
**Confidence:** 4

**Summary:**

The authors of this study investigate the problem of  planning with partial observation, which is important in embodied AI. To achieve this, the authors propose a video generation model Generative World Explorer (Genex), that allows an agent to simulate the world through panoramic representation. The authors also propose an imagination-driven POMDP framework, where generated images assist the agent in decision-making through question-answering (QA).

**Strengths:**

1. Importance of the work: While world models with front-view and multi-view videos are actively investigated by the research community, the generation of panoramic videos is seldom explored. This paper introduces a novel training strategy specifically for panoramic video generation, contributing valuable insights for the community.
2. In this work, the authors aim to define an embodied agent with belief revision driven by imagination, which is able to imagine hidden views through imaginative exploration..
3. Two new datasets called Genex-DB and Genex-EQA have been collected to facilitate the proposed pipeline. The scenarios include a diverse range of styles: Realistic, Animated, Low-Texture, and Geometric.
4. On the proposed dataset Genex-DB and Genex-EQA, the proposed method Genex achieves favorable results in panoramic video generation and embodied QA, compared to other baselines.

**Weaknesses:**

1. My main concern is the **actual impact** of the proposed 'imagination' on embodied QA. While the authors show an approach to link the panoramic video generation with embodied QA, the experiments do not explicitly demonstrate the effectiveness of the ''imagination generation''. How about the results of POMDP without imagination？
2. As far as the reviewer knows, most of the generation models (including the SVD used in this paper) are poor in the reasoning ability, because essentially they are just simulating the probability of objects appearing. In most cases, if there are no explicit constraints like specified object category, the generation model wouldn't expect an ambulance to be here in most cases. This is an open question and the reviewer wants to see the point of the authors. Additionally, could the authors provide more examples of the imagination results, particularly challenging cases like those shown in Fig.12?
3. Some **concerns about the Genex-EQA questions**. The questions and answers in the dataset are quite subjective. For example in the second row and second column in Fig.12, the gt choice is "Signal the car to stop for the pedestrian". This action seems impractical for an autonomous driving vehicle. Further clarification on the methodology and rationale behind the question and answer collection process is needed to understand the dataset's reliability.
4. For panoramic video generation task, though the method serves as a baseline, it is beneficial to have **some comparisons with previous single-view world models** (because they can also perform panoramic video generation task by just replacing the data) and demonstrate why these models fail to generate panoramic videos.

**Questions:**

1. Some typos: L96: imaginatively-imaginative; L186: a-an; Fig.15 Imaginatin-Imagination.
2. Fig link: L379: Fig.2? Maybe this should be Fig.6.

The reviewer has identified four major concerns and would like the authors' responses to these points. Please answer each concern in the rebuttal stage. The reviewer will respond according to the authors' rebuttal in the discussion phase.

---

> ### Author Response · Authors · 2024-11-24
> **Response to Reviewer GZnb (1/3)**
>
> We value the reviewer’s constructive feedback and want to provide additional clarification.
>
> ---
>
> > **W1)** Actual impact of imagination on embodied QA
>
> **In the designated Genex-EQA, POMDP without imagination is impractical and unreliable.**
>
> There are three possible forms of exploration in Genex Embodied QA (Genex-EQA):
>
> - Physical Exploration (POMDP without imagination)
> - Imaginative  Exploration (Genex)
> - No Exploration
>
> _Physical vs. Imaginative Exploration_: **In our EQA benchmark and even in most  real-life situations, physical exploration—or POMDP without imagination—is largely infeasible.** When making emergent decisions in situations requiring immediate responses, such as navigating a sudden traffic obstruction, agents cannot change their physical positions. Moreover, physical exploration is time-consuming and resource-intensive, whereas imagination allows agents to reason instantaneously. For example, in the case of avoiding the ambulance, the ambulance has driven away when the agent physically explores the scene. As a result, **physical POMDPs without imagination are impractical and cannot be achieved in our Genex-EQA benchmark**, but Genex serves the same purpose as physical exploration.
>
> _Imaginative vs. No Exploration_: Our results show that without any form of exploration, single-agent accuracy reaches only 43.50%, while multi-agent accuracy drops further to 21.88%. These figures demonstrate the inability of GPT agents to perform effectively without mental exploration, as they lack the capacity for abstract reasoning or scenario simulation, given only visual and contexture input. When imagination is introduced through Genex, single-agent accuracy dramatically increases to 95.44%, and multi-agent accuracy improves to 94.87%. This substantial improvement underscores the critical role of imagination in enabling agents to reason and make informed decisions under physical constraints.
> | Model                          | Single-Agent Accuracy | Multi-Agent Accuracy |
> |--------------------------------|------------------|-----------------|
> | Multimodal GPT-4o              | 43.50            | 21.88           |
> | **Genex (GPT4-o)**             | **95.44**        | **94.87**       |
>
> These findings highlight the impact of imagination on agent performance in Genex-EQA, particularly when physical exploration is not an option.
>
>
>
> ---
>
> > **W2)** Generation models (including the SVD used in this paper) are poor in the reasoning ability,
>
> We appreciate the reviewer’s insightful observations. We fully agree that most generation models, including the SVD used in our work, are indeed limited in reasoning ability and primarily simulate the probability of objects appearing. This aligns with our own settings, which we consider in our benchmark setting. Specifically, we ensured that in all cases, objects were at least partially observed to avoid scenarios where the model would need to make purely uninformed guesses.
>
> For instance:
> - In _Figure 4 (top)_, the model receives the front view of a stop sign and predicts its back view. This serves as an anchor for the model to infer novel views within the scene.
> - In _Figure 4 (bottom)_, the model project from a fully observed perspective to what another agent might perceive under partial observation. Genex is used to approximate the other agent’s belief while giving its full understanding of the environment.
>
> This careful benchmark setup reflects our agreement with the reviewer’s point and avoids scenarios where the model is required to reason without sufficient observational input.

---

> > ### Author Response · Authors · 2024-11-24
> > **Response to Reviewer GZnb (2/3)**
> >
> > > **W3)** Some concerns about the Genex-EQA questions. The questions and answers in the dataset are quite subjective.
> >
> > We appreciate the reviewer's feedback and acknowledge that the figures presented may involve some subjective interpretations. We made two efforts to make the Genex-EQA more objective:
> > (1) In our original submission, we made the question choices including very specific reasons, to isolate the subjective factor.
> > (2) In our new efforts, we introduce a control set for every scenario.
> >
> > First, we would like to clarify that our Genex-EQA answer choices are specific (Figure 12 choices are abbreviated due to the formatting), such as
> >
> > - Signal the car to stop for the pedestrian _because they are likely to collide_.
> > - Stay in place and wait for the green light _because the it is safe for every agent_.
> > - Honk to alert the pedestrian of the approaching car _because they are moving too fast_.
> > - Proceed cautiously while monitoring both the car and pedestrian _because the path ahead is clear_.
> >
> > The italicized parts (reasons behind the choices) are also included in the given answer choices.
> >
> > This ensures the agent decides with more logical accuracy and removes subjective opinions. In addition, at the original setup, we found if we do not provide such reasons behind the choices, GPT agents tend to make very safe choices, (e.g. always stop in place regardless of the given observation, largely due to their safety protocols in training.), thus providing the reasons alway make evaluation easier. We will make sure this is clarified in the writings.
> >
> > Second, we also introduced a control group for each scenario. For instance, in a case where an agent needs to avoid an ambulance, we included a corresponding case where no ambulance is present. This approach allows us to isolate the impact of specific factors on agent performance and provides a clearer evaluation of their decision-making abilities.
> >
> > The new results are as follows:
> >
> > | Method                  | Decision Accuracy (%) |                | Gold Action Confidence (%) |                | Logic Accuracy (%) |                |
> > |-------------------------|------------------------|----------------|----------------------------|----------------|---------------------|----------------|
> > |                         | Single-Agent          | Multi-Agent    | Single-Agent              | Multi-Agent    | Single-Agent       | Multi-Agent    |
> > | Random                  | 25.00                 | 25.00          | 25.00                     | 25.00          | -                  | -              |
> > | Unimodal Gemini-1.5     | 30.56                 | 26.04          | 29.46                     | 24.37          | 13.89              | 5.56           |
> > | Unimodal GPT-4o         | 27.71                 | 25.88          | 26.38                     | 26.99          | 20.22              | 5.00           |
> > | Multimodal Gemini-1.5   | 46.73                 | 11.54          | 36.70                     | 15.35          | 0.00               | 0.00           |
> > | Multimodal GPT-4o       | 46.10                 | 21.88          | 44.10                     | 21.16          | 12.51              | 6.25           |
> > | **Genex (GPT4-o)**      | **85.22**             | **94.87**      | **77.68**                 | **69.21**      | **83.88**          | **72.11**      |
> >
> > The new results show a lower overall accuracy but still consistent improvements by Genex to confirm Genex's effectiveness. We hope this contributes to a more objective evaluation framework.
> > Additionally, we are expanding the benchmark to include a broader range of objective scenarios and welcome any suggestions and collaborations for further improvement.

---

> > > ### Author Response · Authors · 2024-11-24
> > > **Response to Reviewer GZnb (3/3)**
> > >
> > > > **W4)** comparisons with previous single-view world models
> > >
> > > | **Model**         | **Input**    | **FVD ↓** | **MSE ↓** | **LPIPS ↓** | **PSNR ↑** | **SSIM ↑** |
> > > |--------------------|-------------|-----------|-----------|-------------|------------|------------|
> > > | **→ direct test**  |             |           |           |             |            |            |
> > > | CogVideoX          | six-view    | 4451      | 0.30      | 0.94        | 8.89       | 0.07       |
> > > | CogVideoX          | panorama    | 4307      | 0.32      | 0.94        | 8.69       | 0.07       |
> > > | SVD                | six-view    | 5453      | 0.31      | 0.74        | 7.86       | 0.14       |
> > > | SVD                | panorama    | 759.9     | 0.15      | 0.32        | 17.6       | 0.68       |
> > > | **→ tuned on Genex-DB** |      |           |           |             |            |            |
> > > | Baseline           | six-view    | 196.7     | 0.10      | 0.09        | 26.1       | 0.88       |
> > > | Genex w/o SCL      | panorama    | 81.9      | 0.05      | 0.05        | 29.4       | 0.91       |
> > > | Genex              | panorama    | **69.5**  | **0.04**  | **0.03**    | **30.2**   | **0.94**   |
> > >
> > >
> > > We introduce additional comparison results for different model across three key directions:
> > >
> > > 1. _Panorama vs. Six-View Generation:_
> > >    We compare panorama generation with generation in six separate views. Details of the implementation can be found in Figure 17. While the individual faces in the six-view generation achieve acceptable quality, the egocentric context is lost across views. This lack of shared environmental context leads to worse overall performance metrics in the table.
> > >
> > > 2. _Performance Before and After Training on Genex-DB:_
> > >    For SVD, we evaluate performance on single-view videos (before training on Genex-DB) and on panoramic videos (after training on Genex-DB). Without specific training on panoramic data, SVD struggles with out-of-distribution inputs, often producing random pixels and static frames. CogVideoX, though capable of maintaining panoramic representations, fails to meet our task requirements. It generates static positions with changing objects, but our task requires panoramic navigation, which it cannot achieve.
> > >
> > > 3. _Training with vs. without SCL:_
> > >    Training with SCL leads to a notable 15% improvement compared to training without SCL, demonstrating the importance of SCL in enhancing model performance.
> > >
> > > In addition, our task targets panoramic movement frozen in time. **Existing world models often focus on freezing the current world position and predict change in the next world state, whereas Genex focus on freezing world state but predict change in world position.**
> > >
> > > As a result, current video generation models either fail at generating panoramas or are not designed for the required movement, highlighting the limitations of current approaches.
> > >
> > > If you have other single-view world models (or video generation models) you'd like us to compare, please let us know, and we will make sure to include them in our evaluations.
> > >
> > > ---
> > >
> > > Thank you for your very detailed review of our draft and correction on our typos. We will make sure they are corrected in the revised version.
> > >
> > > ---
> > >
> > > If this does not fully address your concerns, we would appreciate further elaborations.

---

> > > > ### Comment · Reviewer_GZnb · 2024-11-25
> > > > **Official Comment by Reviewer GZnb**
> > > >
> > > > I thank the authors for the response.
> > > >
> > > > Though most of my concerns are addressed, I am still concerned about the impact of imagination. The objectives of generative tasks and planning tasks differ fundamentally, leading to potential conflicts when integrated. Generative tasks aim to predict the most likely future scenarios, optimizing for probabilistic accuracy and often favoring high-probability events. In contrast, planning tasks prioritize safety and robustness, requiring the model to account for low-probability but high-risk scenarios that could have severe consequences. This divergence can result in suboptimal planning performance if the generative task’s outputs are directly used as intermediate representations. A more reasonable approach (to the best knowledge of the reviewer) is to leverage features extracted from the generative model rather than relying on its explicit predictions. This ensures that the planning model maintains its focus on safety-critical decision-making while benefiting from the contextual insights provided by the generative task.
> > > > However, this is still an unexplored problem and does not affect the great contribution of this work to the community if the codes and data are released.
> > > >
> > > > Since my rating is already positive, I will keep a score of 6.
> > > >
> > > > Additionally, regarding W4, I suggest that the authors include a comparison with prior works on world models for autonomous driving, such as VISTA[1] and MagicDrive[2].
> > > > [1] Gao, Shenyuan, et al. "Vista: A Generalizable Driving World Model with High Fidelity and Versatile Controllability." arXiv preprint arXiv:2405.17398 (2024).
> > > > [2] Gao, Ruiyuan, et al. "Magicdrive: Street view generation with diverse 3d geometry control." arXiv preprint arXiv:2310.02601 (2023).

---

### Official Review · Reviewer_MnVr · 2024-11-03

**Soundness:** 3
**Presentation:** 3
**Contribution:** 2
**Rating:** 6
**Confidence:** 3

**Summary:**

The paper works on the problem of decision-making in the partial observation setting. To tackle the task, the authors introduce a novel panorama-based video diffusion model which can imagine the observations from different positions. The authors further combine the generative model and the LLM to help the decision making process. To evaluate the decision making performance, they design a benchmark over 200 scenarios in single and multi-agent settings. The results show that their pipeline achieves better performance by augmenting the agent's imagination ability via the generative model.

**Strengths:**

1. Leveraging generative models to complete the partial observations to a full “world” understanding is reasonable to utilize the priors learned from the data.
2. For the panorama representation, they design the spherical-consistent learning during their learning process to improve the consistency of the panorama image. From their results, the panorama truly shows better consistency and leads to better representation of the scene.
3. The authors conduct extensive experiments and create a benchmark for demonstrating the challenging cases under the partial observation constraints.

**Weaknesses:**

1. In this work, the authors actually construct an explicit representation for “the imagination prior” to make decision making. However, in the benchmark setting, most questions seem only related to a specific case. For single-agents, just try to avoid some unseen cars. And for multi-agent, try to make the other two agents avoid collision. The task setting seems not challenging and common enough to demonstrate the usefulness of such imagination ability. Also it’s hard to see the real performance through such discrete choice-making decision accuracy. Potentially, the method can serve as a role to generate the bird-eye map from a single panorama image and can reveal the hidden cars not in the observation.
2. The paper mentions such imagination can be further updated based on new observations, however, in this work, there is no integration of the imagination and the new observations.

**Questions:**

1. How to determine what’s the trajectory to explore if the world is unlimited? And how to make sure the information is enough to make a decision?
2. Is there better way to evaluate the imagination ability, like the 3D concept error with GT (there is hidden car or not, how much unobserved information is discove

---

> ### Author Response · Authors · 2024-11-22
> **Response to Reviewer MnVr (1/2)**
>
> We appreciate the reviewer’s constructive feedbacks and would like to provide additional justification, clarification, and experimental results.
>
> ---
>
> > **W1)**  The task setting seems not challenging and common enough to demonstrate the usefulness of such imagination ability.
>
> **Not Challenging Enough**
>
> We believe the challenges in our task setting arise from two key aspects: the necessity of imagination and its effectiveness in making decisions.
>
> In our human study, participants were asked whether they relied on their imagination to answer questions based on text and image inputs. More than two-thirds of the successful respondents confirmed that imagination was essential—something that cannot be achieved using a single image alone.
>
> Additionally, Genex helps humans make decisions. Specifically, we observed improvements in human performance from 91.50% to 94.00% in single-agent scenarios and from 55.24% to 77.41% in multi-agent scenarios. This shows that imagination is not purely intuitive or vague. For example, humans may speculate about what others see but often lack the ability to fully reconstruct the occlusions, perspectives, and detailed views of other agents. The strength of Genex lies in its familiarity with the domain of exploration and hence better than humans in constructing different views. These results confirm the difficulty of solving EQAs without Genex, as the task demands reconstructing unseen details and forming actionable representations. Without Genex, both humans and AI agents face significant limitations in inferring occlusions and perspectives, making accurate decision-making challenging.
>
> By combining these factors, our results show a significant improvement in GPT-4o's performance, rising from 44% to 95% in single-agent settings and from 22% to 95% in multi-agent scenarios. This highlights GPT-4o's initial limitations in the designed scenario and the substantial improvements brought by Genex.
>
> **Not Common Enough**
>
> This work establishes a foundation and a starting point for exploring imaginative reasoning in complex scenarios. We demonstrate that such a model is plausible, achieves zero-shot transferability, and provides clear benefits to humans and agents in outdoor scenarios. We are actively working to scale this approach to more complex indoor environments and broader applications for Embodied tasks. We welcome ideas to expand this work further.
>
> ---
>
> > **W1)**  Potentially, the method can serve as a role to generate the bird-eye map from a single panorama image and can reveal the hidden cars not in the observation.
>
> Thank you for your insightful comment. We appreciate your suggestion that our method could generate bird's-eye view (BEV) maps from a single panoramic image to reveal hidden cars. While we considered 360° free navigation at the beginning, we opted to fix the z-axis for egocentric planar exploration to align with our focus on grounded embodied navigation. However, we agree with the potential of extending this method to BEV map generation (which comes with the exact same pipeline) and have conducted preliminary explorations in this direction: [Anonymous GitHub link](https://anonymous.4open.science/r/Genex-Bird-Eye-View/bird-eye.md). This capability enables the agent to imagine a third-person perspective through BEV maps, supporting more informed and objective decision-making. We value your suggestion and would be open to further collaboration on this exciting future research. We will also add the demo to the updated version.
>
> ---
>
> > **W2)** The paper mentions such imagination can be further updated based on new observations, however, in this work, there is no integration of the imagination and the new observations.
>
> Since our imaginative exploration is controlled by a large multimodal model, integrating new observations is straightforward. New observations can be incorporated as additional inputs in a multi-hop conversational format for the LMM, enabling it to control Genex with this new information. The model can then adjust its belief and incorporate it back into the conversational loop. This capability leverages the inherent flexibility of large multimodal models to process and integrate diverse streams of information dynamically, and it is closely tied to the current LMM's long-visual-context processing ability. Our primary aim is to showcase the potential of imaginative exploration and how LMMs can control and reason with it, while incorporating new observations will become more seamless as LMM capacities continue to improve.

---

> ### Author Response · Authors · 2024-11-22
> **Response to Reviewer MnVr (2/2)**
>
> > **Q1)** How to determine what’s the trajectory to explore if the world is unlimited? And how to make sure the information is enough to make a decision?
>
> We use large multimodal models (LMMs) such as GPT-4 for navigation and path planning, where exploration trajectories are determined by prompting the LMM agent with a chain of thought reasoning process. This iterative prompting guides the agent to prioritize areas that maximize information gain while aligning with the task objective. A simplified system pipeline is provided in Figure 15, and an example of an exploration prompt is shown in Figure 14.
>
> Decision-making based on the collected information is also handled by the LMM. After exploring, the model evaluates the gathered observations to update its belief state and make decisions. If the information is deemed insufficient, the LMM is prompted to continue exploring until it determines that enough information has been gathered to make a confident decision. This feedback-driven approach ensures adaptability in complex and potentially unlimited environments, allowing the system to dynamically balance exploration and decision-making based on the task requirements.
>
>
> ---
>
>
> > **Q2**) Is there better way to evaluate the imagination ability, like the 3D concept error with GT (there is hidden car or not, how much unobserved information is discovered)
>
> As we are leveraging image-to-video transformation, determining whether a purely hidden car exists might not be a realistic evaluation metric. For instance, if a hidden car is completely unseeable from the input image, generating such a car would require the diffuser to make a purely speculative guess. For applications requiring safe decision-making, relying on such speculative imagination may introduce risks. Instead, our approach prioritizes evaluating how Genex reconstructs and extrapolates partially observed regions based on the input image, which aligns better with the model's strengths and intended use cases.
> Currently, we are comparing 3D reconstruction models to evaluate Genex’s ability to imagine the novel 3D concept (Section 5.2; Figure 9; Table 3), focusing on how it generates unseen parts of an object. This evaluation tests its ability to construct partially observed objects, confirming it successfully extrapolates and fills in missing information based on the given input.
>
> | Model            | LPIPS↓ | PSNR↑  | SSIM↑ | MSE_obj.↓ | MSE_bg.↓ |
> |-------------------|--------|--------|-------|-----------|----------|
> | TripoSR          | 0.76   | 6.69   | 0.56  | 0.08      | -        |
> | SV3D             | 0.75   | 6.63   | 0.53  | 0.08      | -        |
> | Stable Zero123   | 0.50   | 14.12  | 0.57  | 0.07      | 0.06     |
> | **Genex**             | **0.15** | **28.57** | **0.82** | **0.02** | **0.00** |
>
> ---
>
> We greatly value your suggestions. If any concerns remain, we would appreciate further clarifications.

---

> > ### Comment · Reviewer_MnVr · 2024-11-24
> >
> > Thanks for the responses from the authors. It's great to see the potential of extending existing method to generate "full" BEV map, which can be further used to a better reference for the LMM decision or some other decision-making methods. The role of LMM is to guide the navigation and make the final decison. In the navigation period, it's still hard to see its advantage over some simple frontier based exploration, if want to demonstrate its efficiency, the action step (turn left, move specific distance) actually limits its efficiency. For the decision-making, the benmark is used to show the performance of LMM on different manually constructed scenarios. However, the benchmark is only designed for the specific scenarios needing imagination. In the real usage, it's still an open question on when to trigger such imagination process and if the imagination process will break the normal driving logic in common scenarios. I will raise my score for now for the potential of this work to imagine of full state driving world, however, hope the authors add more analysis on the LMM performance for the navigation efficiency and decision-making influence on normal driving scenarios.

---

> > > ### Author Response · Authors · 2024-12-01
> > >
> > > Dear Reviewer,
> > >
> > > Thank you for your thoughtful feedback. We have updated the manuscript to include the bird's-eye view in Appendix A.10. Thank you again for your suggestions.
> > >
> > > > the action step (turn left, move specific distance) actually limits its efficiency
> > >
> > > Indeed, including action steps adds complexity. However, this highlights the innovation of our method. While simpler frontier-based exploration methods achieve similar functionalities, Genex not only includes their capabilities but also introduces a new degree of freedom in action selection, enabling it to tackle more challenging scenarios effectively.
> > >
> > > > However, the benchmark is only designed for the specific scenarios needing imagination.
> > >
> > > We would like to clarify that this specific focus is intentional. We acknowledge that such designed cases may not occur frequently, but when they do, imagination becomes indispensable. In these situations, the absence of this capability could lead to severe outcomes, such as collisions with pedestrians or other vehicles. This highlights the crucial role of imagination in addressing these high-stakes scenarios effectively.

---

### Official Review · Reviewer_M4Hx · 2024-11-04

**Soundness:** 3
**Presentation:** 3
**Contribution:** 4
**Rating:** 8
**Confidence:** 3

**Summary:**

The paper introduces the challenge of planning with partial observation in embodied AI and highlights how humans can mentally explore unseen parts of the world to update their beliefs and make informed decisions. To replicate this human-like ability, the authors propose the Generative World Explorer (Genex), a video generation model that enables agents to mentally explore large-scale 3D worlds and acquire imagined observations to update their beliefs. They train Genex using a synthetic urban scene dataset, Genex-DB, and demonstrate that it can generate high-quality and consistent observations during long-horizon mental exploration and improve decision-making in an existing model.

**Strengths:**

+The idea of build a Generative World Explorer is interesting, and I think it will be useful to the development of embodied AI research.

+ It's practical to apply the proposed Genex to the embodied decision making process.

**Weaknesses:**

-There is a gap between the training data (synthesized with unity) and test data (captured from google street), the degrees of freedom of in the observation perspectives, google street seems to more limited compared to the unity. But the gap between training and test data may not  be always "bad", because such gap may show more "Generalizability".

-In the following sentence “An embodied agent is inherently a POMDP agent (Kaelbling et al., 1998): instead of full observation, the agent has only partial observations of the environment.” , “a POMDP agent” seems to lack rigor. POMDP (Partially Observable Markov Decision Process) is a modeling framework that can be applied to describe the behavior of an agent in an environment where full state information is not available. Visual observation is only one channel for information acquisition. Saying that incomplete visual observation necessarily leads to a POMDP is also not very rigorous.

**Questions:**

Overall I think this is a good paper that can contribute to the subsequent development of the field of embodied AI.

---

> ### Author Response · Authors · 2024-11-20
> **Response to Reviewer M4Hx**
>
> We thank the reviewer for their insightful comments and for appreciating our work.
>
> ---
>
> > **W1)** There is a gap between the training data and test data.
>
> We also agree with the reviewer that while there are differences in testing environments, this gap provides a valuable opportunity to evaluate the model's generalizability, which is critical for real-world applications.
>
> In terms of cycle consistency, the diffuser achieves a consistency score (latent MSE) as low as 0.07 when trained and tested on different data within the same scene. Additionally, it maintains a consistency score of approximately 0.1 for cross-scene, demonstrating impressive zero-shot generalizability to real-world scenarios. This zero-shot capability highlights its potential for future real-world applications.
>
>
> | trained on Realistic (UE5), generalized to other scenes (engines) | **Realistic** (UE5)  | **Real-World** (Google Map Street View) | **Indoor** (BEHAVIOR Vision Suite) | **Anime** (Unity) | **Texture** (Blender) | **Geometry**  (Blender) |
> |------------------------|------------|------------|---------------|-----------|-------------|--------------|
> |  **Mode**  | train-test  | zero-shot     | zero-shot            | zero-shot    | zero-shot      | zero-shot       |
> | **Exploration Cycle Consistency**  | 0.068  | 0.105     | 0.091            | 0.124    | 0.133      | 0.2047       |
>
>
> The columns are by testing environment.
>
> ---
>
> > **W2)** On the use of a POMDP agent – Saying that incomplete visual observation necessarily leads to a POMDP is also not very rigorous.
>
> We acknowledge the reviewer's concern regarding the phrasing in our manuscript and appreciate them pointing it out. POMDP is indeed a modeling framework that describes scenarios with partial observability, with visual observations being only one modality of information. While our intention was to emphasize that the agent operates under partial observability, we agree that the phrasing could be more precise. We will revise this statement to better reflect the framework's scope and ensure clarity.
>
> ---
>
> If there is anything else you'd like us to address, please let us know.

---

### Official Review · Reviewer_GPuR · 2024-11-05

**Soundness:** 2
**Presentation:** 2
**Contribution:** 2
**Rating:** 5
**Confidence:** 3

**Summary:**

Humans have the capacity to imagine the future and revise their beliefs about the world based on these imagined observations. Building on this concept, the authors have introduced a video generation model, GeNex, which enables an agent to mentally explore future imagined observations. Subsequently, a Large Language Model (LLM) is utilized as the policy model to predict future actions.

**Strengths:**

The concept is intriguing and the explanation is clear and straightforward.

**Weaknesses:**

1. There are some errors in the mathematical formulations presented, particularly in equations (3) and (4), which are confusing.
2. Although SCL is highlighted as a contribution, its effectiveness is not demonstrated in the experimental results.
3. The use of latent diffusion with temporal attention is not a novel architecture.
4. The real-world dynamics of vehicles do not allow for pure rotation, which the paper seems to overlook.
5, Table 3 presents an unfair comparison.

**Questions:**

1. Equation (3) is incorrect
2. The derivation of Equation (4) is unclear. Could you explain how it was formulated?
3. Is the LLM policy model fine-tuned or used as is?
4. The space of 'state' & 'belief' is not clearly defined.
5. It is unclear whether the diffusion model has been overfitted to the dataset, potentially making it inadequate for handling complex real-world interactions.
6. The entire framework appears to have little connection with POMDP.

---

> ### Author Response · Authors · 2024-11-18
> **Response to Reviewer GPuR (1/2)**
>
> We appreciate the detailed feedback from the reviewer. We would like to add additional clarification.
>
> > **W1) Q1) Q2)** There are some errors in the mathematical formulations presented, particularly in equations (3) and (4), which are confusing.
>
> $
> \text{(Equation 3)} \quad b^{t+M}(s^{t+M}) = \prod_{t}^M  \bigg( \underbrace{O(o^{t+1} | s^{t+1}, a^t) \sum_{s^t} T(s^{t+1} | s^t, a^t)}_{\text{Physical Exploration}} \bigg) b^{t}(s^{t})
> $
>
>
> Equation (3) follows the standard POMDP formulation [Kaelbling et al., 1998](https://people.csail.mit.edu/lpk/papers/aij98-pomdp.pdf) (page 107). In our variation, we introduce a multiplication of exploration steps to account for a sequence of **physical exploration**. The timestep $t$ in our model represents an **exploration sequence**. This modification allows us to encapsulate iterative exploration within a single belief update.
>
> If there are specific aspects of our formulation that appear incorrect, we would appreciate further clarification to address them appropriately.
>
> - - -
>
> $\text{(Equation 4)} \quad \hat{b}^{t}(s^{t}) = \prod_{i}^I  \bigg( \underbrace{ p_{\theta}(\hat{o}^{i+1} | o^i, \hat{a}^i ) }_{\text{Imaginative Exploration}} \bigg) b^{t}(s^{t})$
>
> Equation (4) is derived by replacing the traditional physical exploration components of the POMDP belief update with an imaginative exploration mechanism driven by a diffusion-based generative model parameterized by $\theta$. In the standard belief update (Equation 3), the agent transitions between states $s^t$ using the transition model $T(s^{t+1} | s^t, a^t)$ and incorporates actual observations $O(o^{t+1} | s^{t+1}, a^t)$, which requires summing over all possible prior states to account for uncertainty. By contrast, in **imagination-driven belief revision**, the agent remains in the current state $s^t$ and employs the diffusion model $p_{\hat{\theta}}(\hat{o}^{i+1} | \hat{o}^i, \hat{a}^i)$ to generate a sequence of hypothetical observations based on imagined actions $\hat{a}^i$. This substitution eliminates the need for state transitions and the associated summations because the physical state does not change; instead, the belief is updated multiplicatively by the probabilities of the imagined observations across the imaginative steps $I$. As a result, the belief $\hat{b}^t(s^t)$ is directly refined by the product of these generated observation probabilities applied to the initial belief $b^t(s^t)$, leading to Equation (4). This approach leverages the generative capabilities of the diffusion model $\theta$ to simulate potential observations, enabling the agent to perform instantaneous and iterative belief revisions without altering the underlying state, thereby enhancing the efficiency and flexibility of the belief update process.
>
>
> > **W2)** Although SCL is highlighted as a contribution, its effectiveness is not demonstrated in the experimental results.
>
>
> We compare the performance of the SVD diffuser trained on Genex-DB dataset, both with and without SCL, on the same dataset.
>
> | Model        | FVD ↓  | MSE ↓  | LPIPS ↓ | PSNR ↑  | SSIM ↑  |
> |----------------|--------|--------|---------|---------|---------|
> | w/o SCL            | 81.9   | 0.05   | 0.05   | 29.4    | 0.91  |
> | **w/ SCL**      | **69.5** | **0.04** | **0.03** | **30.2** | **0.94** |
>
> The results demonstrate that SCL enhances video quality, achieving a **15% improvement** in the FVD metric.
>
> Furthermore, for exploration cycle consistency, we observe greater improvements as the number of generation steps increases during generative exploration.
>
> | Model  / # Generation Step            | 2  | 3   | 5 | 10  |
> |----------------|--------|--------|---------|---------|
> | w/o SCL      | 0.070 | 0.079  | 0.105 | 0.197 |
> | **w/ SCL**            | **0.067**  | **0.061**    | **0.069**    | **0.081**  |
> | improvement            | 4.3%  | 22.8%    | 34.3%    | 58.9%  |
>
> As generation step occurs, edge inconsistencies accumulate over multiple generations, causing the input image to become increasingly out-of-distribution for the diffuser. This ultimately leads to a decline in exploration generation, where **training with SCL keeps the image generation in-domain of the diffuser over many inferences**.
>
>
>
> > **W3)** The use of latent diffusion with temporal attention is not a novel architecture.
>
> We would like to clarify that our work does not emphasize the use of temporal attention as a core contribution. Our model is grounded in SVD, which we found sufficient for our task. The discussion of latent diffusion with temporal attention is included solely to explain the referenced work.  Temporal attention is a module from SVD, which we have appropriately cited. This aspect was not intended to highlight our own contributions. We will revise the text to ensure clarity and accuracy in this regard.

---

> > ### Author Response · Authors · 2024-11-18
> > **Response to Reviewer GPuR (2/2)**
> >
> > > **W4)** The real-world dynamics of vehicles do not allow for pure rotation, which the paper seems to overlook. 5, Table 3 presents an unfair comparison.
> >
> > We would like to clarify the context of Table 3 and address the concern about pure rotation.
> >
> > Table 3 is intended to evaluate the generation quality of Genex by comparing it to other novel view synthesis methods. In this experiment, we place an object in the scene, use Genex to simulate forward movement, and evaluate the generated observation of this object from a new perspective (e.g., generating a high-quality back view given a front view). This process is an essential aspect of creating a coherent and realistic generated world. **All the compared models are specifically designed and trained for cyclic or rotational novel view generation. This experiment does not involve views captured from a vehicle’s perspective.** Could you elaborate on what makes this comparison "unfair"? We would be glad to provide additional clarification or further details regarding any concerns about this comparison.
> >
> > In addition, one of the main motivations for using a panorama-based representation is its capacity for pure rotation, which significantly facilitates world exploration. **While real-world vehicle dynamics do not allow for pure rotation, Genex’s effectiveness is highlighted by its ability to overcome this limitation with unlimited rotation and navigation.** This enables agents to fully observe their surroundings, supporting more robust decision-making.
> >
> > Finally, our work is not limited to navigation from the perspective of a vehicle. **Genex is capable of all embodied scenarios**, enabling imaginative exploration from the observation of a person, a car, or any other agent.
> >
> >
> >
> >
> >
> > > **Q3)** Is the LLM policy model fine-tuned or used as is?
> >
> > In our experiments, the policy model is used as is, without any fine-tuning.
> >
> > > **Q4)** The space of 'state' & 'belief' is not clearly defined.
> >
> > The state is the environment the agent is currently situated in, and the space is the entire world. As described in Equation (4), we remove the transition of states, simplifying the definition. At the beginning, the state is represented as a 3D environment used to sample the initial observation.
> > The belief operates at a higher level and pertains to the LLM's internal reasoning. Through continuous prompts in multi-hop conversations, the language model continually revises its internal belief about the world. These beliefs are encoded within the model's internal parameters, evolving as new observations and prompts are processed. Additionally, if we ask the agent to explicitly state its belief, the space would be represented in natural language.
> >
> >
> > > **Q5)** It is unclear whether the diffusion model has been overfitted to the dataset, potentially making it inadequate for handling complex real-world interactions.
> >
> > **We have conducted extensive experiments to evaluate the generalizability of Genex.** The numerical results are presented in Section 5.2 (Table 2), and the visual demonstrations are included in Appendix A.7 (Figure 18). Our results indicate that Genex, trained on synthetic data, demonstrates robust zero-shot generalizability to real-world scenarios. Specifically, the model trained on synthetic data performs well on scenes such as indoor behavior vision suites, outdoor Google Maps Street View in real-world settings, and other synthetic scenes that all differ significantly from the training distribution, without additional fine-tuning.
> >
> >
> >
> > > **Q6)** The entire framework appears to have little connection with POMDP.
> >
> > While our framework does not strictly adhere to the traditional POMDP formalism, it fundamentally builds upon its core principles. Specifically, the state in our framework corresponds to the agent's environment, while the belief represents the agent’s internal reasoning and its evolving understanding of the world based on observations. Unlike standard POMDPs, which require physical exploration of the environment to update beliefs and gather new information, we replace that component with Genex. By enabling mental simulation and navigation, Genex streamlines the belief-updating process, significantly reducing the time and resource demands of physical exploration. This abstraction integrates reasoning and decision-making within complex, unstructured environments, fully leveraging and extending the foundational ideas of POMDPs.
> >
> >
> >
> > If this does not fully address your concerns, we would appreciate further elaborations.

---

> > > ### Author Response · Authors · 2024-11-24
> > >
> > > Dear Reviewer,
> > >
> > > Thank you for your feedback. We provided our response five days ago and hope it addresses your questions. Please let us know if there are any remaining concerns or points, and we will do our best to clarify everything before the rebuttal period ends.
> > >
> > > Thank you!

---

> > > > ### Comment · Reviewer_GPuR · 2024-11-24
> > > >
> > > > I've increased the score to 5. If the authors can further explain the mentioned concerns, I would further increase the score.

---

> > > > > ### Author Response · Authors · 2024-11-26
> > > > > **Reply to Reviewer GPuR (1/2)**
> > > > >
> > > > > We thank the reviewer’s feedback and would provide further clarification. We appreciate the opportunity for further explanation.
> > > > >
> > > > >
> > > > > > In my point of view, this paper seems to employ a world model and LLM to provide pseudo labels for the final policy model, which is not a new idea to me.
> > > > >
> > > > >
> > > > > - We would like to clarify that our approach explicitly distinguishes between observations and world states, grounded in the physical world. In contrast, the cited works (e.g., Hao et al.) operate under the assumption that observations directly equate to world states, predicting future world states directly. However, in real-world settings, observations are inherently partial and do not fully represent the underlying world state. We highlight this challenge and emphasizes our aim to address it by developing agents that imaginatively explore their environment to update their beliefs about the world state.
> > > > > - We would like to clarify the distinction in problem formulations between world models and our approach. The world models are designed to **predict the _future_ world states**, while we formulate our problem to **gain more complete observations of the _current_ world states**.  The key distinction lies in their purpose: Genex offers diverse perspectives of the same scene simultaneously to enhance the agent's understanding of the present, whereas existing world models focus on predicting future scenarios to aid in forecasting.
> > > > >
> > > > > We appreciate the opportunity to elaborate on these distinctions and hope this explanation provides clarity.
> > > > >
> > > > > Hao et al. (also cited in our manuscript) employ _"a world model that predicts the next state of reasoning after applying an action to the current state,"_ which fundamentally differs from our approach. Their method assumes a **complete understanding of the current world state**, with the world model providing feedback on **how an action modifies that state in the next timestep**. In contrast, our work addresses scenarios where the agent **lacks a full understanding of the current world state (i.e., received partial observation)**. Here, the agent performs imaginative actions—mental simulations that do not result in real-world consequences—to **explore and revise their belief about the current world state, all while keeping the present state unchanged (i.e. freeze the time)**.
> > > > >
> > > > > For instance, when driving with a limited view, a standard world model predicts what will happen in the next second if the current trajectory continues—for example, estimating the car's position after moving forward in time. In contrast, our model freezes the current moment in time (everything in the environment freezes their movements) and provides alternative perspectives of the same state, such as visualizing the scene from different angles. This fundamental difference—predicting future changes versus deepening understanding of the present—highlights how our approach helps agents form a more complete and immediate grasp of their environment and is distinct from previous approaches.

---

> ### Comment · Reviewer_GPuR · 2024-11-24
>
> I appreciate the authors' detailed responses, which addressed most of my initial concerns.
>
> For Q1, about the formula error, I've checked P107 of the book, the formula follows
> $b'(s')= $
> $ \frac {O(s',a,o)\sum _ {s\in s}\left(T(s,a,s')b(s)\right)}{Pr(o|a,b)} $ .
>
> while in your paper, this function becomes,
> $b'(s')= $
> $ \frac {\left(O(s',a,o)\sum _ {s\in s}T(s,a,s')\right)b(s)}{Pr(o|a,b)} $ .
>
> It is clearly mistaken and wrong. Eq(3) should contain a matrix multiplication term in order to expand multi-step exploration. Could the authors please explain this misalignment step by step？Another issue is that the normalizer is neglected in most of the equations.
>
> For Eq(4), it should be a random sampling process in order to marginalize the $o^i$ and $\hat{a}^i$ variables (for example, monte carlo Tree Search). The original formula make me very confusing. The future video generation is a complex transition distribution instead of a deterministic process. How does this model handle complex future frame prediction if the dangerous vehicle is completely unobserved?
>
> In my point of view, this paper seems to employ world model and LLM to provide pseudo labels for the final policy model, which is not a new idea to me.
>
> Note that some previous papers already expressed similar ideas: world model for reasoning.
> Reasoning with Language Model is Planning with World Model. Hao et al. (abstract: RAP repurposes the LLM as both a world model and a reasoning agent, and incorporates a principled planning algorithm based on Monte Carlo Tree Search for
> strategic exploration in the vast reasoning space. During reasoning, the LLM (as agent) incrementally builds a reasoning tree under the guidance of the LLM (as world model) and rewards, and efficiently obtains a high-reward reasoning path with a proper balance between exploration vs. exploitation.)

---

> ### Author Response · Authors · 2024-11-26
> **Reply to Reviewer GPuR (2/2)**
>
> > **Equation Misalignment**
>
> For equation (3), a key distinction between our formulation and that of Kaelbling et al., 1998 lies in the timing of belief updates. **In the original POMDP formulation, the belief is updated at every step as the agent takes actions and receives observations**, for continuous refinement of the belief state. However, in our design, the exploration path is preplanned, meaning that all actions and observations during the exploration phase are fixed ahead of time. This allows us to treat exploration as a separate module, isolating it from the belief update process. As a result, **for our formulation, the belief is only updated after the entire exploration phase is completed, reflecting the cumulative insights gathered during exploration**.
>
> In POMDP, the belief is defined as:
>
> $
> b'(s') = \frac{O(s', a, o) \sum_{s \in S} \left( T(s, a, s') b(s) \right)}{Pr(o \mid a, b)},
> $
>
> where $b'(s')$ is the updated belief state, $O(s', a, o)$ represents the observation probability, $T(s, a, s')$ is the state transition probability, and $Pr(o \mid a, b)$ is the normalizing factor.
>
> When exploration spans multiple steps ($M$ steps), the agent accumulates observations and transitions over these steps. Belief updates occur only after this phase is complete. Below, we present the belief updates for $M = 1$ and $M = 2$, leading to the general $M$-step formulation:
>
> 1. **For $M = 1$:**
>
> $
> b^{t+1}(s^{t+1}) = b^t(s^t) \cdot \left( O(o^{t+1} \mid s^{t+1}, a^t) \sum_{s^t} T(s^t, a^t, s^{t+1}) \right).
> $
>
> 2. **For $M = 2$:**
>
> $
> b^{t+2}(s^{t+2}) = b^t(s^t) \cdot \left( O(o^{t+1} \mid s^{t+1}, a^t) \sum_{s^{t+1}} T(s^t, a^t, s^{t+1}) \right)
> \cdot \left( O(o^{t+2} \mid s^{t+2}, a^{t+1}) \sum_{s^{t+1}} T(s^{t+1}, a^{t+1}, s^{t+2}) \right).
> $
>
> 3. **General Form for $M$-Steps:**
>
> $
> b^{t+M}(s^{t+M}) = b^t(s^t) \cdot \prod_{k=1}^{M} \left( O(o^{t+k} \mid s^{t+k}, a^{t+k-1}) \sum_{s^{t+k-1}} T(s^{t+k-1}, a^{t+k-1}, s^{t+k}) \right).
> $
>
> Here, $t$ represents the time step within the exploration phase, starting from the initial time $t$ and progressing sequentially up to $t+M$.
>
>
> The **Physical Exploration** term, $\sum_{s^t} T(s^{t+1} \mid s^t, a^t)$, models transitions between states during the exploration process. This, combined with the observation term $O(o^{t+1} \mid s^{t+1}, a^t)$, ensures that all contributions to the belief are accounted for before the final update.
>
> By structuring belief updates in this way, exploration is a reasoning phase that aggregates information over multiple steps. The preplanning of exploration paths enables this modularity, and belief updates only occur once the process is complete. This modular approach allows the agent to process the effects of exploration comprehensively, reflecting the accumulated transitions and observations in the belief state at the end of the exploration phase. Equation (3) formalizes this process.
>
> ---
>
> With that being said, we fully understand your concern regarding the placement of the belief term. If you believe it is better to adhere strictly to the original POMDP setup, we would adjust the formulation to place the belief term appropriately within the process to align with standard conventions.
>
> ---
>
> Regarding the normalization term, indeed, for simplicity, we removed it in the current formulation. We will ensure that it is reintroduced in the revision for preciseness and to maintain consistency with the standard POMDP formulation. We appreciate the reviewer for pointing this out.
>
>
> ---
>
> > **Equation (4)** The future video generation is a complex transition distribution instead of a deterministic process.
>
> As discussed earlier, we do not handle future events. The world state is fixed, with all dynamics in the current world halted. Instead of simulating state transitions, we generate new observations to complete the agent’s partial view of the world. Therefore, there is no state transition in our approach.
>
> $\text{(Equation 4)} \quad \hat{b}^{t}(s^{t}) = \prod_{i}^I  \bigg( \underbrace{ p_{\theta}(\hat{o}^{i+1} | o^i, \hat{a}^i ) }_{\text{Imaginative Exploration}} \bigg) b^{t}(s^{t})$
>
> In Equation 4, the action $\hat{a}^i$ is explicitly marked with a hat to indicate that it has no real-world consequences. The world, along with everything in it, remains unchanged. While the video generation model introduces randomness, as reflected in $p_{\theta}$’s probabilistic nature, the underlying world state remains frozen, making it neither distributional nor deterministic.
>
> ---
>
> If this does not clarify your concerns, we would like to provide further explanation.

---

> > ### Author Response · Authors · 2024-12-02
> >
> > Dear Reviewer,
> >
> > We have provided our reply five days ago. If our response does not address your remaining concerns, please let us know, and we will address them promptly before the rebuttal period concludes. In addition, we have revised our manuscript based on your suggestions.
> >
> > Thank you!

---

> ### Comment · Reviewer_GPuR · 2024-12-02
>
> > In contrast, our work addresses scenarios where the agent lacks a full understanding of the current world state (i.e., received partial observation). Here, the agent performs imaginative actions—mental simulations that do not result in real-world consequences—to explore and revise their belief about the current world state, all while keeping the present state unchanged (i.e. freeze the time).
>
> If I synthesize a new perspective view for a partially-occluded object that I saw in the previous frames, can this be called unobserved? I don’t think so. This is only `unobservable` to the perception model, but it is `fully-observable` for people. For objects that are fully unobserved in the previous frames, such as objects in occluded areas, it is obviously difficult to synthesize a corresponding new perspective using the method proposed in the paper. For example, for dangerous objects in occluded areas, how to use the method mentioned in the paper to model them to improve the safety.
>
> So I feel that this paper is a bit like novel-view enhancement to improve perception capbility for LLM, and it has little to do with the POMDP and partial observation model mentioned in the paper. It will be interesting if the authors can provide some examples for fully unobservable objects, and it will make the proposed method more applicable to realistic world.

---

> ### Comment · Reviewer_GPuR · 2024-12-02
>
> > However, in our design, the exploration path is preplanned, meaning that all actions and observations during the exploration phase are fixed ahead of time. This allows us to treat exploration as a separate module, isolating it from the belief update process. As a result, for our formulation, the belief is only updated after the entire exploration phase is completed, reflecting the cumulative insights gathered during exploration.
>
> I fully understand the exploration phase is fixed. However, in reality, the exploration observation should give a complex high-dimensional distribution, instead of a fixed result, due to the novel-view uncertainty.
>
> I believe a better way is like this: By modeling novel-view exploration as a distribution using SVD, we can sample from SVD with different seeds to model multiple possibilities. Each sampling process can be seen as a Monte-Carlo sampling process. The overall belief updates can be computed by multiple MC samples.
>
> If the exploration results are fixed, I believe you cannot call the initial states `unobservable`, since all the outcomings can be observed using our SVD model.
>
>
> > The conversion from original POMDP to the Equation in $M=1$,
> $b^{t+1}(s^{t+1})=b^t(s^t)\cdot (O(o^{t+1}|s^{t+1},a^t) \sum_{s^t}T(s^t, a^t, s^{t+1}))$
>
> Apparently, in POMDP formulation, $s^t$ has multiple possible values, for example $s^t=s_1$, $s^t=s_2$, $s^t=s_3$. Since $s^t$ is not fully observed, it can be a complex distribution.
>
> $b^{t+1}(s^{t+1})= O(o^{t+1}|s^{t+1},a^t) \sum_{s^t}T(s^t, a^t, s^{t+1})b^t(s^t)$
>
> $= O(o^{t+1}|s^{t+1},a^t) (T(s_1, a^t, s^{t+1})b^t(s^t=s_1) + T(s_2, a^t, s^{t+1})b^t(s^t=s_2) +T(s_2, a^t, s^{t+1})b^t(s^t=s_2) )$
>
> Apparently, it cannot be simplified as,
>
> $b^{t+1}(s^{t+1})=b^t(s^t=?)\cdot (O(o^{t+1}|s^{t+1},a^t) (T(s_1, a^t, s^{t+1}) + T(s_2, a^t, s^{t+1}) + T(s_2, a^t, s^{t+1}))$
>
> The only case in which the formula in the paper is correct is that there is only one kind of state for $s^t \in \{{s_{fixed}}\}$. However, it becomes a fully observable problem, since there is only one state. So the formulation of Eq(3)&Eq(4) makes me very confusing.
>
>
> > As discussed earlier, we do not handle future events. The world state is fixed, with all dynamics in the current world halted. Instead of simulating state transitions, we generate new observations to complete the agent’s partial view of the world. Therefore, there is no state transition in our approach.
>
> As discussed earlier, if the states are partial observed, there are unlimited possibilities for new observations. If there is only one possibility for new observations, then the initial state should be fully observed (at least with the helf of SVD). So I still think the formulation in the paper is a little bit confusing.
>
> The results in this paper is interesting, but I still think the formula in the paper should be improved to make it more clear to understand.

---

> ### Author Response · Authors · 2024-12-02
> **Reply to Reviewer GPuR (1/2)**
>
> Dear Reviewer,
>
> Thank you for your reply!
>
> ---
>
> > So I feel that this paper is a bit like novel-view enhancement to improve perception capability for LLM.
>
> We agree that our work shares similarities with novel-view enhancement. Different from standard novel-view techniques, our approach enables unlimited spatial exploration in all dimensions. This allows for possibilities beyond static viewpoint synthesis and further control. Moreover, since the exploration process—though facilitated by diffusion generation—is controlled by an agent model, it mirrors a POMDP structure akin to real-world physical exploration (we agree with your points about uncertainty and would clarity below).
>
> > If I synthesize a new perspective view for a partially-occluded object that I saw in the previous frames, can this be called unobserved?
>
> In our work, we primarily focus on objects that are never fully observed from any angle or perspective. For example, an object might remain consistently occluded or partially visible throughout all frames, leaving certain parts completely unviewed and requiring synthesis based on incomplete information. We also acknowledge the reviewer’s point that completely unobserved objects or scenarios inherently involve uncertainty due to the nature of the diffusion process. Because of this, our work centers on safe decision-making scenarios, specifically focusing on partially observed scenes.
> We also see the potential of the cases involving the generation of entirely new objects or environments, we provide sample demonstrations in our anonymous video demo: www.youtube.com/watch?v=rFwuCTsrYVU (e.g., partially observed scenarios at 1:25 ,completely unobserved cases at 0:35, zero-shot generation at 1:00). While time constraints during the rebuttal period limit our ability to provide extended exploration examples for completely unobserved scenes, we plan to include such examples in future revision.

---

> > ### Author Response · Authors · 2024-12-02
> > **Reply to Reviewer GPuR (2/2)**
> >
> > > However, in reality, the exploration observation should give a complex high-dimensional distribution, instead of a fixed result, due to the novel-view uncertainty. I believe a better way is like this: By modeling novel-view exploration as a distribution using SVD, we can sample from SVD with different seeds to model multiple possibilities. Each sampling process can be seen as a Monte-Carlo sampling process. The overall belief updates can be computed by multiple MC samples. If the exploration results are fixed, I believe you cannot call the initial states unobservable, since all the outcomings can be observed using our SVD model
> >
> > We appreciate your clarification and now understand your concern. We resonate with your comments and fully agree with your points. Our previous response aims to clarify that the current physical state of the world is fixed and certain, and indeed the imaginative exploration is uncertain and not fixed. Rest assured, in line with your perspective, our video generation models (based on SVD) produce non-deterministic predictions from a probabilistic distribution.
> >
> > We minimize the discussion of generation variation primarily because the partially observed nature limits result variability. We acknowledge your point and further believe that longer-range exploration introduces greater uncertainty. We can think about two examples:
> > - Assuming we want to explore a narrow alley in long-range. From our initial observation, we can see a door in the alley (from this perspective, the door appears as a narrow line). The outcome of the exploration is uncertain—the door could either be open or closed.
> > - If our exploration involves a person and their hands are obscured, the state of their hands—whether raised or lowered—remains uncertain during the exploration process. The model will sample one of the possible outcomes.
> > The above long-range & articulation examples will be our future research works.
> >
> > Formulating the problem as a model with multiple possibilities from the outset—such as using a Monte Carlo sampling process, as you suggested—would improve precision and clarity. We appreciate your feedback on fixed vs. varied results and the POMDP formulation and will revise the manuscript accordingly.
> >
> > ---
> >
> > Thank you for your comment. We believe it will be instrumental in improving the quality of our work.

---

> > > ### Author Response · Authors · 2024-12-03
> > >
> > > Dear Reviewer,
> > >
> > > We aim to discuss and address all your concerns before the rebuttal period concludes. Please let us know if this clarifies your concern.
> > >
> > > Thank you.

---

### Author Response · Authors · 2024-12-04
**Summary of the discussion period**

Dear Reviewers and Chairs,

We sincerely thank all reviewers for their constructive feedback and for recognizing the contributions of our work.

We have carefully addressed all concerns and provided detailed responses, along with additional experiments to support our claims:

- Clarified multiple points about the experimental setup and included model comparisons for `M4Hx`.
- Expanded on the generalizability of the proposed model in response to reviewer `M4Hx`.
- Addressed reviewer `MnVr`'s concerns regarding the subjectiveness of the benchmark and practical applications of Genex, highlighting its future potential.
- Conducted additional experiments with single-view world models to resolve reviewer `GPuR`'s feedback.

We believe these updates adequately address all concerns raised. Although we have not yet heard back from reviewer `GPuR`, we hope our revisions meet their expectations.

Thank you again for your valuable feedback and engagement.

---

### Meta-Review · Area_Chair_ps2M · 2024-12-23

**Metareview:**

**Summary**

The paper proposes Generative World Explorer (Genex), which uses a video generation model to imagine taking a sequence of actions and their followup observations.  The imagined observations are used to revise agent beliefs of the world, which are then used for decision making.  The framework can be extended to multi-agents so that decisions can be made while taking into account other agent's beliefs.

To demonstrate the feasibility of the approach, the work uses a diffusion-based video generation model that generates egocentric equirectangular projected panoramas.  For decision making, a large multimodal model (LMM) is used for the policy model and for mapping observation to belief (the LMM is prompted to select actions based a set of input images).

A dataset (Genex-DB) of rendered panoramic images from four scenes with different styles is used to train video diffusers (one per scene).  The dataset also contains additional indoor and outdoor panoramic images for testing.  Genex-DB is used to evaluate the video generator.  In addition, an embodied QA dataset (Genex-EQA) consisting of 200 scenarios is constructed to evaluate the decision making ability of the proposed model.

The main contributions of this work are 1) the proposed Genex framework for using imagining outcomes of different actions and integrating the imagined observations with an LMM for decision making  2) the egocentric panoramic diffuser, 3) Genex-DB and Genex-EQA datasets

**Strengths**

Reviewers noted the following strengths of the work:
1. Idea of generative world explorer is interesting [GPuR,M4Hx,GZnb]
2. Generation of panoramic video is underexplored [GZnb] and spherical consistent learning seems effective [MnVr]
3. Contributed dataset and benchmarks [GZnb,MnVr]

**Weaknesses**

Reviewers have concerns about the following:
1. Limited experiments
   - Limited comparison to previous single-view world models [GZnb]
   - Experiments do not actually show impact of imagination [GZnb] / EQA questions doesn't really seem to require imagination [MnVr]
2. Limited integration of imagination and new observations [MnVr]
3. Concerns about the quality of the Genex-EQA dataset [GZnb]
4. Concerns about mathematical formulations [GPuR] and connection of the proposed framework to POMDP [GPuR,M4Hx]
5. Concerns about generalizability of trained diffusers [GPuR]

**Recommendation**

Given the overall positive rating from reviewer (3 vote for accept, one for reject), the AC believe that the contributions of the work could be interesting for the ICLR audience.  The AC finds the experimental setup somewhat weak.

As pointed out by several reviewers, it's unclear how much imagination (vs just exploration the environment) is required for the proposed Genex-EQA benchmark.  There is also limited information about the quality of the Genex-EQA benchmark.

Nevertheless, the AC believes the contributions (proposed framework with panoramic video diffuser) is sufficiently interesting for acceptance at ICLR.

**Suggested updates**

The AC notes that the submission can be improved with the following:
1. Add discussion to clarify relation to POMDP and clearly explain how the Equation 3 samples trajectories through exploration.
2. Provide additional detail on how Genex-EQA is created.  Appendix 4 only consists of A.4.1 (Dataset details) which did not have much information about how Genex-EQA was constructed and how the quality of Genex-EQA was ensured.

**Additional Comments On Reviewer Discussion:**

The paper initially received scores of 3 [GPuR], 5 [MnVr], 6 [GZnb], 8 [M4Hx].  Reviewer GPuR expressed some concerns about the mathematical formulation presented by the work and the precise relation to POMDP, as well as whether the proposed spherical-consistent learning (SCL) was effective.   During the author response period, the reviewer's concerns were partially addressed (the authors provided ablation showing the effectiveness of SCL, and explained the difference in their formulation to standard belief estimates that are updated after every action) and R-GPuR increased their score to 5 (still marginally negative).

Reviewer MnVr and GZnb had questions about the impact of imagination in the proposed benchmark (Genex-EQA) and the difficulty and quality of the benchmark, as well as some other concerns.  For R-MnVr, some of their questions were addressed during the author response period, and they increased their rating from 5 to 6.  Reviewer GZnB kept their rating at 6.  Reviewer M4Hx was very positive on the work, but their review did not provide much useful information.

---

### Decision · Program_Chairs · 2025-01-22

Accept (Poster)